# Wetropolis extreme rainfall and flood demonstrator: from mathematical design to outreach

Onno Bokhove[1], Tiffany Hicks[1], Wout Zweers[2], and Thomas Kent[1]

[1]School of Mathematics and Leeds Institute for Fluid Dynamics, University of Leeds, LS2 9JT, Leeds, UK
[2]Wowlab, Enschede, The Netherlands

**Correspondence:** Onno Bokhove (o.bokhove@leeds.ac.uk)

**Abstract.** Government and consulting experts on flood mitigation generally face difficulties when trying to explain the science of extreme flooding to the general public, in particular the concept of a return period. Too often, for example, people perceive they are safe for the next 100 years after a $1:100$ year return-period flood has hit their town. UK flood practitioners therefore gave us the challenge to design an outreach tool that conceptualises the science of flooding in a way that is accessible to and directly engages the public, and in particular demonstrates what a return period is. Furthermore, we were tasked with designing a live 3D physical model rather than a graphical or animated 2D game on a screen. We show here how we tackled that challenge by designing, constructing and showcasing the 'Wetropolis flood demonstrator'. Wetropolis is a transportable and conceptual physical model with random rainfall, river flow, a flood plain, an upland reservoir, a porous moor, representing the upper catchment and visualising groundwater flow, and a city which can flood following extreme and random rainfall. A key novelty is the supply of rainfall every Wetropolis day. Several aspects of Wetropolis are considered:

(i) We present the modular mathematical and numerical design on which Wetropolis is based. It guided the choice of parameter values of Wetropolis, which was loosely inspired by the Leeds' Boxing Day floods of the River Aire in 2015. The design model further serves as building block and inspiration for adaptations suited to particular local demands. Moreover, the model is purposely lean and therefore quick to compute, serving flexibility in the outreach-tool design, but is less suitable for any detailed scientific validation.

(ii) The constructed Wetropolis is described here in broad terms but we include a link to a GitHub site with details to inspire other bespoke designs. The goal, again, is to facilitate new adaptations of Wetropolis for particular catchments different to the Leeds' River Aire case.

(iii) Our experience in showcasing Wetropolis is summarised and discussed, with the purpose to give an overview as well as inspire improved and bespoke adaptations. While Wetropolis should be experienced live, with videos found on the GitHub site, here we provide a photographic overview. To date, Wetropolis has been showcased to 500 to 1000 people at public workshops and exhibitions on recent UK floods, as well as to flood practitioners and scientists at various research and stakeholder workshops.

(iv) We conclude with some ongoing design changes, including how people can experience natural flood management in a revised Wetropolis design. Finally, we also discuss how Wetropolis, although originally focussed solely on outreach, led to a

new cost-effectiveness analysis and protocol for assessing flood-mitigation plans as well as inspired other physical models for use in education and water management.

## 1   Introduction

The Boxing Day flood of 2015, and more recently Storm Ciara in 2020, caused widespread damage in Yorkshire, UK, due to extreme flooding of the River Aire in and around Leeds and the River Calder in and around Todmorden, Mytholmroyd and Hebden Bridge. Thankfully no fatalities occurred but the economic damage was severe and estimated to be around £500M (West Yorkshire, 2016). November 2015 was the third wettest month on record in terms of precipitation and December 2015 was the wettest on record (Environment Agency, 2016). As such, the soil was already saturated when $48$hrs of extreme rainfall

fell in Yorkshire, leading to the Boxing Day floods of 2015. After this and other recent floods, many flood victims asked why extreme flood events, seemingly occurring more and more often, were causing such havoc in their communities.

   To provide some mathematical background on modelling, mitigation and statistics of extreme flood events, we were asked to disseminate scientific background on "risk in the age of extremes" at the citizens' conference "Science of floods" in Hebden Bridge (Science of floods, 2016). In the first two decades of the 2000s, including the Boxing Day and Storm Ciara floods,

Hebden Bridge was hit by both summer flash-floods and winter floods leading to concerns amongst flood victims that their lives and properties were insufficiently protected. It led to further and intense discussions with environmental agencies on the need for more and different types of flood defences. Important questions in this discussion are the following:

   – Is it going to rain more in the future in the UK?

   – Can we define extreme precipitation and flooding events?

– How (well) can we predict heavy precipitation and floods?

   – Finally, how can we elucidate these questions, their answers and uncertainties, in an interactive table-top demonstration? The latter question was posed to us by experts on flood mitigation, in their wish to communicate better with the general public. In particular, experts faced troubles explaining what a return period is in flooding, alongside general difficulties explaining statistical notions involved in flood mitigation.

We will discuss some answers to these questions in turn given their relevance to Wetropolis' design.

   *Is it going to rain more in the future in the UK?*  Both the IPCC report (IPCC, 2013) and Sanderson's UKCP09 report of the Met Office (Sanderson, 2010) show that there is no increase of significance in average annual rainfall foreseen in climate projections, not across the globe on average and also not in the UK. However, there are geographical and seasonal variations foreseen: winter rainfall will generally increase and summer rainfall will thus decrease but with more intense downpours.

*Can we define extreme precipitation and flooding events?* Extreme events tend to be expressed as the chance that an extreme event occurs on one day in a year. If that chance is $1\%$, for example, then we say that this extreme event has a return period of $1:100$ years. Flooding events can be classified in terms of such return periods as, e.g., $1:10, 1:20, 1:50$ and $1:100$ or $1:200$ year events, with the latter two considered to be extreme. The uncertainty in an event with a $1:100$ year return period will be larger than one with a $1:20$ year return period. An extreme event occurs in the tail of a probability distribution and may have never been observed. Events with return periods longer than the data record cannot be classified directly from that data and must be determined using theoretical probability distributions. However, the low number of extreme events in a finite-time data set means it is difficult to establish accurately the tail of the distribution. Accordingly, there is a great deal of uncertainty associated with extreme events, which should ideally be quantified and communicated effectively. By assuming a suitable (parametric) probability distribution function (pdf) one can use the data to fit the parameters of the pdf and subsequently generate data ("sample the pdf") in the tail for events with return periods beyond the length of the data record. Classical pdfs, such as (half a) Gaussian distribution or a Gamma distribution, are typically used to model rainfall intensities but are not suitable for extreme events. Extreme-value theory offers a family of distributions to overcome the limitations of the classical pdfs. Given a sufficient amount of (rainfall or river level) data over a certain threshold, the Generalised Pareto distribution (GPD) attempts to model the probabilities of extreme events beyond the data record. It is also possible to capture both the bulk and the extreme tail of the distribution in a so-called mixture model by combining a Gamma and a Generalised Pareto distribution (GPD), cf. Wong et al. (2014) The above is relevant because within Wetropolis we will use a "discrete" distribution with a "rare" tail.

*How (well) can we predict heavy precipitation and floods?* The Boxing Day floods of 2015 were caused by large-scale winter rainfall. The $48$ hours of consecutive rainfall in the days leading up to the flood were the wettest on record: in Bradford $69.4$mm and in Bingley $93.6$mm of rainfall were measured over $48$ hours. We will use this direct response of floods driven by one or two days of heavy rainfall in our design. It resulted in the flooding of the River Aire with river-level records reached in Leeds and elsewhere along the river. In Armley, Leeds, the gauge station measured a maximum river level of $5.22$m while the previous electronic record was $4.03$m in the Autumn of 2000 (Environment Agency, 2016). The river level during the 1866 flood was roughly around $4.5$m, cf. Bokhove et al. (2018a) (their Fig. 3). Both rainfall and river levels were by and large well-predicted by the UK Met Office via Numerical Weather Prediction and by the Environment Agency (–EA, cf. a presentation by an EA– Yorkshire leader in Leeds). Predictions are generally quite good for large-scale winter rainfall and the resultant changes in river levels. Downpours, e.g. in the summer, tend to be more localised and are therefore much more difficult to predict in terms of location, intensity and duration. The same holds for resulting flash-floods and downpour induced surface-water flooding. Hence, simply put, fluvial or river flooding in winter tends to be easier to predict than pluvial or surface-water flooding events in the summer.

*Finally, how can we elucidate these questions, their answers and uncertainties, in an interactive table-top demonstration?* The decision to design a table-top demonstrator was triggered by the desire of flooding experts to have a 3D physical set-up as opposed to 2D animations or computer graphics on a screen. It follows a recent trend to use transportable physical set-ups,

such as the coastal wave tank of JBA Trust (an online video has attracted to date over 5.7M views)[1]. The expression of extreme events in terms of return periods is difficult to grasp and often misunderstood, especially by the public. The Boxing Day flood of 2015 was classified as an event with a $1:200^+$ year return period – including the unclear meaning of the plus sign in $200^+$ (Environment Agency, 2016)[2]. That does not mean that it has to take another $200^+$ years, so until after 2215, before

5 the next Boxing-Day-type flood might occur in Leeds. It does, however, mean that the average time between events of similar magnitude will be $200^+$ years, given a sufficiently long record of "stationary" statistical data. To let people experience such an extreme event in a table-top set-up they can of course not be asked to wait for 200 years on average, so our design for a flood demonstrator with rainfall must be scaled down both in size and duration. Miniature river flooding has been demonstrated in small-scale experiments (e.g., as in the Lego model of Pampaloni et al. (2018)) but these all tend to involve *deterministically*

imposed extreme water input –with water inflow supplied and adjusted deterministically and/or manually. The key novelty in our design lies in the way rainfall is supplied *randomly* to our table-top hydrodynamic set-up for both river and groundwater flow. We have modified the classical symmetric Galton board, inspired by such a set-up used at Leeds' School of Mathematics open days. A typical Galton board has a tilted surface in which a (steel) ball falls down under gravity and encounters a series of symmetric pins or channel corners, each determining with a $p$ and $(1-p)$ chance whether the ball continues or falls to the

left or to the right. The design is usually such that $p \approx 1/2$ but small variations can occur in practice and after a series of $n$ rows of splittings a binomial distribution arises, given a sufficiently large number of trials. Moreover, for $n \to \infty$, a Gaussian distribution emerges. The Galton board is often used to visualise and demonstrate statistical distributions in real time during, e.g., outreach events. To obtain an asymmetric discrete distribution with a discrete tail representing relatively-extreme events, the standard symmetric Galton board described above was modified as follows. For $p = 1/2$ and $n = 1$, the first and only split

leads to a $(1,1)/2$–distribution. The first split of the second row for $n = 2$ is now eliminated while the second split is not, leading to a $(3,1)/4$–distribution. Continuing to the third row of splittings as usual, for $n = 3$, we obtain a $(3,4,1)/8$–distribution. The last and fourth row for $n = 4$ yields the final $(3,7,5,1)/16$–distribution. An image of such a asymmetric Galton board is given in Fig. 1. Two of these Galton boards will be used to supply rain to our table-top river and ground-water flow model, one concerning the duration and amount $(0.1, 0.2, 0.4, 0.9)r_0$ of rainfall during a Wetropolis day, with its unit wd, and another one

concerning the location of the rainfall. Rain duration will be either $(10, 20, 40, 90)\%$ of the amount $r_0$ of rainfall per wd and rain location will be either rainfall (i) in a reservoir with generally instant run-off into the river; (ii) in both a reservoir and on a moor; (iii) on the moor with groundwater flow and its nonlinear, delayed release of water into the river; and, (iv) no rain, in the Wetropolis catchment. See Fig. 2 for a plan view of Wetropolis. Both duration and location are determined by the outcome of one trial through two Galton boards per wd, together yielding a 4 by 4–matrix of joint probabilities, with the no-rain case

having a rare chance (rare for the "UK") of $1/16^{\text{th}}$ comprised by four of those 16 outcomes. By design, an extreme event occurs when it rains $90\%$ in both locations (i.e. in the reservoir and moor) with a chance of $7/256 \approx 0.0273 = 2.73\%$, which in our construction will by design lead to flooding of a city further downstream along a (winding) river in the set-up. A Boxing-Day-

---

[1] https://www.youtube.com/watch?v=3yNoy4H2Z-o and https://www.jbatrust.org/how-we-help/physical-models/wave-tank/; this tank was designed by a team, including OB, of the Centre for Doctoral Training in Fluid Dynamics in Leeds, upon a request by JBA Trust.

[2] In a recent personal communication with a Yorkshire flood expert, a $1:300$ year return period was mentioned, with no formal confirmation yet to date.

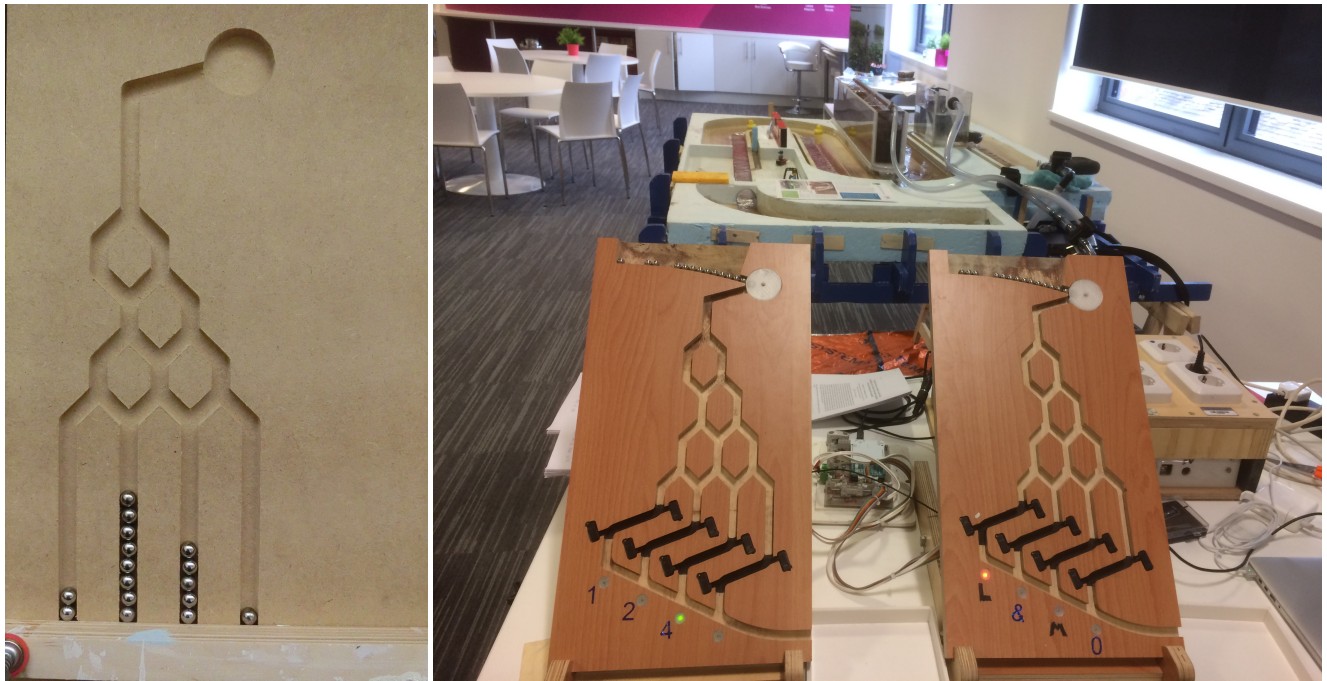

**Figure 1.** Photographs of asymmetric Galton boards. Test board on the left and a final set-up on the right. At every split the chance of a steel ball falling to the left or right is $50\%$ for a well-balanced Galton board. When a sufficiently large number of steel balls falls through this Galton board, the discrete distribution becomes $(3, 7, 5, 1)/16$. The $4 \times 4$ possible outcomes in two of such boards, registered in each by four electronic eyes (located in the black-painted areas along $2 \times 4 = 8$ channels marked here by "$1, 2, 4, \dots$" and "L,&,M,0"), determine both the rainfall amount and its location(s) in Wetropolis. The outcome of the random draw, shown by the lit-up lights, will in this instance lead to $4$s of rain in the lake/reservoir. Photos: OB & WZ.

type event with two consecutive days of extreme $90\%$–rainfall then has a chance of $49/(256^2) \approx 0.000748 = 0.0075\%$. The next and crucial step in the design is to identify and determine the various unknowns in order to ascertain whether a feasible design for a working physical set-up is possible at all.

Given a (winding) river of length $L$ and curvilinear coordinate $s$ along this river, these remaining key unknowns are as follows:

- the influx discharge $Q_0$ at the upstream boundary at $s = 0$;

- the locations $s_{res}$ and $s_m$ where the reservoir and moor enter into the river (with the distance along the river given by a curvilinear coordinate $s$), with a section further downstream along the river comprising a city plain that is prone to (extreme) flooding;

- the rainfall amount $r_0$ (dimensionally a speed, as we will explain in the next section), determining the strength of the pumps required and whether pumping rates can be realistic at all; and,

- the length of a Wetropolis day, in relation to the extreme rainfall and corresponding extreme flooding event, such that the viewer experiences some irritation in having to wait for a randomly-induced extreme event but on average will experience such an extreme event within a reasonable time, i.e. on average within several minutes.

We have chosen $s_{res}$, $s_m$, $Q_0$ a priori and will determine $\mathrm{wd}$ and $r_0$ via visual optimisation of a series of simulations of a simplified mathematical and numerical model (as explained in §2.3). Note that the latter model is a lean design model exclusively geared towards obtaining quick estimates of the design parameters. We emphasise that it is not intended as a predictive model for validation of measured data, but to facilitate and assess design changes efficiently. While the individual components of the mathematical and numerical model are not new in separation, their holistic combination and coupling with our random rainfall delivered by a Galton board is indeed novel. A plan view of a sketch of Wetropolis is given in Fig. 2. The river-canal combination established is inspired by the River Aire and Leeds-Liverpool canal sharing a large part of the same river valley with the canal allowing some minor flood alleviation via (manual) flood control. Further, implicit motivation may stem from the fact that OB and WZ are Dutch citizens and from the context of the Dutch Deltaworks, of which small-scale test versions were built in the "Noord-Oost" (North-East) polder after the 1953 North Sea flood; conceptual modelling of river flood components like in Wetropolis is perhaps natural for Dutch engineers and designers. See also (online) literature regarding the "Waterloopkundig laboratorium" in the "Noord-Oost" polder.

This introduction has given both an anecdotal and scientific background to Wetropolis' inception. Our intention here is to document its journey from design to outreach, and in doing so to inspire and enable readers to redesign Wetropolis bespoke to their own local catchment characteristics; the remainder of the article has the following outline. The above unknowns were determined via visual optimisation of simulations of an idealised mathematical and numerical model of Wetropolis before any design and construction of the table-top set-up were undertaken. This mathematical and numerical modelling is therefore explained in detail and used to determine (some of) the design unknowns in §2. The resulting table-top design of the Wetropolis flood demonstrator is disseminated in §3. Our experience in demonstrating Wetropolis to the general public and to flood practitioners is summarised in §4, including the a priori surprising outcome that professionals in flood prediction and mitigation have also been inspired by Wetropolis, despite that our primary aims have been public engagement. A discussion is found in §5, in which we outline some new designs and future directions, as gathered from such public-engagement sessions.

## 2 Mathematical design

The mathematical model of Wetropolis comprises random rainfall and space-time continuous hydraulic modelling of interconnected river flow, reservoir- and canal-level changes as well as groundwater flow in the moor. While the individual modelling elements in separation are known or straightforward, their holistic combination with the statistical rain modelling as well as the subtle mass-conserving coupling between the elements is nontrivial and new. In addition, dissemination of the model is also required to facilitate adaptations by the readers. One other reason to be quite pedagogical is to reach a wider readership of enlightened and interested members of the public, especially educators. Subsequently, we will establish a numerical discretisation of this space-time continuous model and use numerical simulations to determine the *a priori* unknown parameters of rainfall

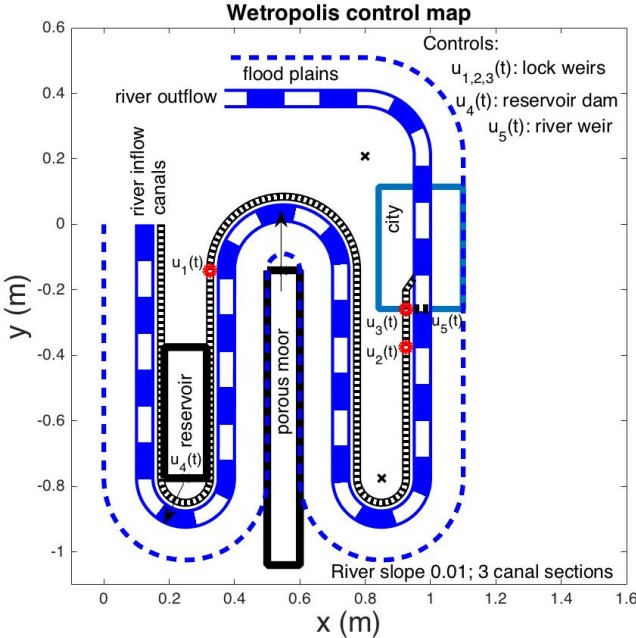

**Figure 2.** Plan view of the Wetropolis table-top experiment. The main river channel is indicated in white-blue blocks and its one-sided flood plain extent by a dashed line. A "Leeds-Liverpool" canal with lock-weirs flanks the $1:100$–sloped river, which has constant upstream inflow and gets fed by water from a reservoir as well as a porous moor filled with lava grains. In both locations, it can rain intermittently and randomly. Outflow is at the end of the river channel, after a city plain that can flood and where the canal flows into the river. Water falling in a full reservoir flows instantly with a manually adjustable fraction of $0 < \gamma \leq 1$ into the canal and the river, the latter with a fraction $(1 - \gamma)$. The reservoir level can also be adjusted manually, which provides some flood control. This control can be adjusted manually to demonstrate the role of a holding reservoir to lessen flooding in cases of extreme rainfall.

amount $r_0$ and wd. Other parameters will be determined heuristically in order to obtain desirable and practical dimensions of the experimental set-up. We present the model here completely; full details of the numerical discretisation can be found in Appendix A.

## 2.1 Statistical modelling of randomised rainfall

As discussed in the introduction, rainfall is modelled stochastically via the outcome of draws from two Galton boards. In the mathematical model these outcomes are simulated, while in the physical flood demonstrator we have either used two actual Galton boards with two steel balls or one Galton board with one steel ball running through two consecutive Galton-board channels. We have discretised rainfall into two categories: location and rain amount, per wd. Rain location has four outcomes: rain in reservoir, moor and reservoir, moor, or no rain in the catchment with a discrete distribution of $(3, 7, 5, 1)/16$. Independently, rain amount has per location four outcomes $(1, 2, 4, 9)r_0/\text{wd}$ with again the discrete distribution $(3, 7, 5, 1)/16$;

**Table 1.** Joint probability matrix of the $4 \times 4 = 16$ outcomes of rainfall times 256, with the extreme case of $7/256$ shown in bold underlined. The rows show P(location), and columns P(amount), with both summing to the imposed discrete distribution $(3, 7, 5, 1)/16$. Since the P(location) and P(amount) are independent, the joint probability is their product.

| | $r_0$ | $2r_0$ | $4r_0$ | $9r_0$ |
|---|---|---|---|---|
| reservoir | 9 | 21 | 15 | 3 |
| both | 21 | 49 | 35 | **7** |
| moor | 15 | 35 | 25 | 5 |
| no rain | 3 | 7 | 5 | 1 |

hence, there are $4 \times 4 = 16$ outcomes determined as a direct product of these two independent distributions, given in Table 1. The possible rain amounts per wd are therefore $(0, 1, 2, 4, 8, 9, 18)r_0$ and the value $r_0$ will be determined in the subsequent modelling such that there is no flooding for $(1, 2, 4)r_0$/wd rainfall, with potentially limited flooding for $(8, 9)r_0$/wd and generally major flooding in the city plain for $18r_0$/wd. The resulting distribution of the rainfall per wd in the 'Wetropolis catchment' (to be read with Table 1) therefore becomes:

$$0r_0: \quad P(\text{no rain}) = 1/16, \tag{1a}$$

$$1r_0: \quad P(1r_0)P(\text{reservoir}) + P(1r_0)P(\text{moor}) = 9/256 + 15/256 = 24/256, \tag{1b}$$

$$2r_0: \quad P(1r_0)P(\text{both}) + P(2r_0)P(\text{reservoir}) + P(2r_0)P(\text{moor}) = 21/256 + 21/256 + 35/256 = 77/256, \tag{1c}$$

$$4r_0: \quad P(2r_0)P(\text{both}) + P(4r_0)P(\text{reservoir}) + P(4r_0)P(\text{moor}) = 49/256 + 15/256 + 25/256 = 89/256, \tag{1d}$$

$$8r_0: \quad P(4r_0)P(\text{both}) = 35/256, \tag{1e}$$

$$9r_0: \quad P(9r_0)P(\text{reservoir}) + P(9r_0)P(\text{moor}) = 3/256 + 5/256 = 8/256, \tag{1f}$$

$$18r_0: \quad P(9r_0)P(\text{both}) = 7/256. \tag{1g}$$

A pdf of this discrete distribution for a computer trial over 500wd's is shown in Fig. 3 (blue bars) with the expected values overlaid (crosses). Suitable values for $r_0$ and wd are established by further modelling, described next.

## 2.2 Mathematical modelling in space-time

The key components of Wetropolis are a river channel with a one-sided flood plain, a groundwater moor, a reservoir, and canals with three segments separated by lock-weirs. The canal flows into the river in the city plain which lies at the downstream end of the set-up. We refer to Fig. 2 for a plan view locating these elements in the actual table-top experiments. In the original design model, the locations of the reservoir and moor have been swapped and we have used a shorter river channel. Simplified mathematical sub-models of these different elements are derived next in isolation before being coupled into one complete and novel mathematical model of rainfall and flooding via suitable boundary and interface conditions.

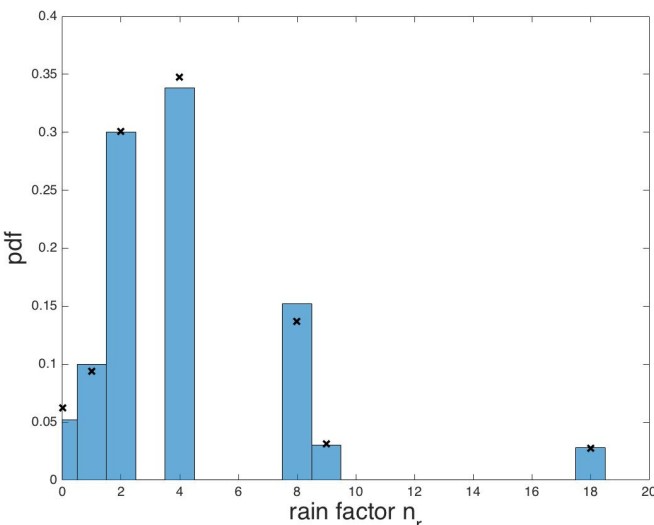

**Figure 3.** The pdf of random rainfall outcomes over 500wd's, displayed as a scaled histogram, is compared with the theoretical discrete pdf (1), denoted by the black crosses. Extreme cases with $18r_0$ rainfall are observed in moor and reservoir combined, here in this 500wd trial with an occurrence on the average.

### 2.2.1 River dynamics

River flow is often modelled as one-dimensional flow in a channel with a cross-section $A = A(s,t)$ as function of space, with a horizontal curvilinear coordinate $s \in [0, L]$ along a winding river channel of length $L$, and time $t$. Both $A(s,t) = A(h; b(s))$ and the in-situ water depth $h = h(s,t)$, above a fixed river bottom $b = b(s)$, and the mean flow velocity $u(s,t)$ are all averaged over the cross section of the river. Hence, the river bottom lies at $z = b(s)$ with vertical coordinate $z$ and the river surface at $z = b(s) + h(s,t)$. E.g., for a rectangular river channel we have $A = w_r h$ with fixed width $w_r$ but in general $A(h; s)$ can depend in a complicated fashion on the river depth $h$ and directly on $s$ – the latter via $b = b(s)$. The governing equations are the Saint-Venant equations (e.g., Bates et al (2010)) consisting of continuity and momentum equations, augmented with the source terms at the locations $s_m$ and $s_{res}$ along the river where the water flows from the moor and reservoir into the river, as well as a parameterization of the channel friction, i.e.

$$\partial_t A + \partial_s(Au) \equiv Q_m \delta(s - s_m) + Q_{res}\delta(s - s_{res}) + Q_{1c}\delta(s - L_{1c}) \tag{2a}$$

$$\underline{\partial_t u + u\partial_s u + g\partial_s h} = -g(\partial_s b + C_m^2 u|u|/R^{4/3}) \tag{2b}$$

with the phenomenological Manning friction coefficient $C_m \in [0.01, 0.15]\mathrm{m}^{-1/3}\mathrm{s}$, cf. Munson et al. (2005), the hydraulic radius $R(h) = \frac{\text{wet area}}{\text{wetted perimeter}}$ (in m), acceleration due to gravity $g$, and the discharge rates $Q_m$, $Q_{res}$ and $Q_{1c}$ of the water flows from the moor, reservoir and canal into the river at locations $S_m, S_{res}$ and $L_{1c}$. Partial derivatives have been denoted by $\partial_s = \partial/\partial s$ and $\partial_t = \partial/\partial t$. These discharge rates are modelled as point sources using delta functions $\delta(s - s_m)$ and $\delta(s - s_{res})$.

In reality the same inflow rates will occur along finite-length, short strips along the river, centred around $s_m$ and $s_{res}$. The two unknown fields are $A$ and $u$ with $h = h(A; s)$ an (often) implicit relation at every location $s$. For the above example with a river channel of rectangular cross-section, we find $R(h) = \frac{w_r h}{2h + w_r}$ and $h = A/w_r$. Initial conditions $A(s, 0)$ and $u(s, 0)$ have to be imposed at $t = 0$ as well as boundary conditions $A(0, t)$, $u(0, t)$ and $A(L, t)$, $u(L, t)$ at $s = 0$ and $s = L$. These latter boundary conditions are used (partially) according to the way the characteristics at $s = 0, L$ of the hyperbolic equations (2) determine whether the boundary data are flowing into the domain or not, cf. Toro (2001).

Even though the actual winding river channel with its one-sided flood plain and city plain has a varying cross-section, for the design calculations we made the simplification to model only a rectangular river channel. In addition, a zeroth-order kinematic model approximation to the Saint-Venant equations (2) has been used for the limiting case with positive velocity $u > 0$ and a constant slope $-\partial_s b > 0$ of the river channel. To zeroth-order, we assume that the bed slope and friction are locally in balance, i.e., the underlined terms in (2) are negligible, such that we obtain

$$u = R(h)^{2/3} \sqrt{-\partial_s b}/C_m. \tag{3}$$

This is a classical approximation used in hydraulics (Munson et al., 2005): the river flow is thus modelled by a kinematic or scalar hyperbolic equation in $A$, as follows

$$\partial_t A + \partial_s (A R^{2/3} \sqrt{-\partial_s b}/C_m) \equiv \partial_t A + \partial_s Q_f(A) = Q_m \delta(s - s_m) + Q_{res} \delta(s - s_{res}) + Q_{1c} \delta(s - L_{1c}), \tag{4}$$

with an upwind information speed $dQ_f(A)/dA > 0$ for flux $Q(A) = Au$ and inflow $A(h(0, t); s = 0)$. Note that the flux $Q_f = Q_f(A)$ is an implicit function of $A$ since $h = h(A)$. For $u > 0$ with $A = w_r h$, it is a kinematic model or nonlinear, scalar conservation law in the water depth $h$ given by

$$\partial_t (w_r h) + \partial_s (w_r h R(h)^{2/3} \sqrt{-\partial_s b}/C_m) \equiv \partial_t (w_r h) + \partial_s Q_f(h) = Q_m \delta(s - s_m) + Q_{res} \delta(s - s_{res}) + Q_{1c} \delta(s - L_{1c}) \tag{5}$$

with the flux $Q_f = Q_f(h)$ rewritten in terms of $h$, initial condition $h(s, 0)$ and upstream influx $Q_0(0, t) = w_r h(0, t) u(0, t)$ defining $h(0, t)$ since $u$ is expressed in terms of $h$ through (3). The Saint-Venant equations are more advanced than the above kinematic model and allow both sub- and supercritical flows. An interim and better model arises when we instead of (3) use the following balance

$$u = R(h)^{2/3} \sqrt{-\partial_s (b + h)}/C_m, \tag{6}$$

which, after substitution into the continuity equation, (2a) yields an advection-diffusion equation, cf. Bates et al (2010). Both these more advanced models are not required for the design estimates but when one wishes to perform (more) accurate predictions of the hydrodynamics in Wetropolis then such advanced models may be required.

### 2.2.2 Groundwater flow

Groundwater levels after rainfall are made visible in a transparent and elongated rectangular box filled to a high level with porous small lava rocks. The box is open at the top and one side, and has walls at the remaining three sides and the bottom.

Rain falls uniformly along this box via a copper pipe with a series of equidistant holes. The groundwater dynamics are modelled to zeroth-order by assuming that the surface of the grains is flat, the rainfall uniform per surface area, that there is no surface run-off and that the fallen rainwater infiltrates sufficiently fast to contribute instantly to the groundwater level $h_m(y,t)$ with coordinate $y$ in a different direction, locally orthogonal to $s$ at the location $s_m$ where the groundwater flows into the river, cf. the delta function in the continuity equation (2a) and kinematic equation (5). A depth-averaged groundwater model with level $h_m(y,t)$ from Barenblatt (1996) is used in a cell of width $w_v$ and length $L_y$, e.g. $w_v = 0.095$m and coordinate $y \in [0, L_y = 0.932$m]. The nonlinear diffusion equation for the groundwater level $h_m(y,t)$, taken to be uniform in the lateral direction, is

$$\partial_t(w_v h_m) - \alpha g \partial_y(w_v h_m \partial_y h_m) = \frac{w_v R_m(t)}{m_{por}\sigma_e} \tag{7}$$

with moor rainfall $R_m(y,t) = R_m(t)$, porosity $m_{por} \in [0.1, 0.3]$, the fraction $\sigma_e \in [0.5, 1]$ of pores filled with water, $\alpha = k/(\nu m_{por}\sigma_e)$ with permeability $k = 10^{-8}m^2$ and viscosity $\nu = 10^{-6}m^2/s$. The boundary conditions are no flux through the wall at $y = L_y$ such that $\partial_y h|_{y=L_y} = 0$, while at $y = 0$ the moor is held at the level $h_3(t)$ of canal–3, the upstream branch of the canal running in parallel to the river, i.e., this a time-dependent Dirichlet condition $h_m(0,t) = h_3(t)$. The mass flux of moor water running in the river (at $s = s_m = 2.038$m) is

$$Q_m(t) = (1-\gamma)Q_{tm} \equiv (1-\gamma)\frac{1}{2}m_{por}\sigma_e w_v \alpha g(\partial_y h_m)^2|_{y=0}, \tag{8}$$

where $Q_{tm}$ is the mass flux following from integration of (7) over the domain $y \in [0, L_y]$ and $0 < \gamma < 1$ the fraction of moor water entering into the river. The reason to multiply by $m_{por}\sigma_e$ is that the water volume in the matrix of particles in the moor changes suddenly from a space filled with pores into free space.

### 2.2.3 Reservoir

The reservoir is a rectangular box of dimensions $h_{res} \times w_{res} \times L_{res}$ with time-dependent water level $h_{res}(t)$, e.g. $L_{res} = 0.293$m and $w_{res} = 0.123$m. In the physical model, the random rainfall enters either via a pipe or a long pipe with numerous holes visualising the rainfall and it leaves the reservoir via an overflow pipe, here modelled simply as a straight weir, cf. Munson et al. (2005). Overflow of the reservoir once it is overfilled is not modelled. The reservoir-level dynamics are governed by

$$w_{res}L_{res}\frac{dh_{res}}{dt} = w_{res}L_{res}R_{res}(t) - Q_{res} \quad \text{with} \quad Q_{res} = C_f\sqrt{g}\,w_{res}\max(h_{res} - P_{wr}, 0)^{3/2}, \tag{9}$$

in which $L_{res}w_{res}$ is the area of the reservoir such that $L_{res}w_{res}h_{res}$ is its time-dependent volume, $P_{wr}$ is the overflow height of the weir, $R_{res}(t)$ the reservoir rainfall and $Q_{res}$ the flux down into the river. Note that the coefficient $C_f$ is dimensionless. The weir is located at $s = s_{res} = 0.925$m, where water flows into the river, cf. the delta function in the continuity equation (2a) and kinematic equation (5).

### 2.2.4 Canal sections

One canal of uniform width $w_c$ runs alongside the river, cf. the Leeds-Liverpool canal and the River Aire in central Leeds, in which the three canal sections are separated by lock-weirs and have mean, time-dependent water depths $h_{1c}(t)$, $h_{2c}(t)$, and

$h_{3c}(t)$. We are thus ignoring currents and height changes along the separate canal sections. Each canal section has a certain depth and is separated from the river by a berm. Canal–3 is the highest and is blocked off on one end, at $s = 0$, and has a weir located at $s = L_{3c} = 1.724$m. Its level is modelled as the variation of its volume due to partial inflow from the moor and outflow of water via a weir in canal–2

$$5 \quad w_c L_{3c} \frac{\mathrm{d}h_{3c}}{\mathrm{d}t} = \gamma Q_{tm} - Q_{3c} \quad \text{with} \quad Q_{3c} = C_f \sqrt{g}\, w_c \max(h_{3c} - P_{3w}, 0)^{3/2} \tag{10}$$

with weir height $P_{3w}$. Canal–2 resides from $s \in [L_{3c}, L_{2c}]$ with $s = L_{2c} = 3.608$m and is modelled likewise but with inflow $Q_{3c}$ from canal–3 and outflow $Q_{2c}$ into canal–1:

$$w_c(L_{2c} - L_{3c}) \frac{\mathrm{d}h_{2c}}{\mathrm{d}t} = Q_{3c} - Q_{2c} \quad \text{with} \quad Q_{2c} = C_f \sqrt{g}\, w_c \max(h_{2c} - P_{2w}, 0)^{3/2}. \tag{11}$$

The section of canal–1 runs from $s \in [L_{2c}, L_{1c}]$ with $L_{1c} = 3.858$m, width $w_c$ and depth $h_{1c}(t)$. It is modelled in the same
10  manner with inflow from canal–2 and outflow $Q_{1c}$ into the river, as follows

$$w_c(L_{1c} - L_{2c}) \frac{\mathrm{d}h_{1c}}{\mathrm{d}t} = Q_{2c} - Q_{1c} \quad \text{with} \quad Q_{1c} = C_f \sqrt{g}\, w_c \max(h_{1c} - P_{1w}, 0)^{3/2}. \tag{12}$$

The weir at $s = L_{1c}$ where water flows into the river is assumed to be subcritical, i.e., we assume there is a sufficient drop from canal–1 to the river level. In terms of height levels, canal–3 has a berm at $z = 0.06$m and its bottom resides at $z = 0.04$m; canal–2 has a berm at $z = 0.04$m and its bottom resides at $z = 0.02$m, and canal–1 has a berm at $z = 0.021$m and its bottom
15  resides at $z = 0.001$m. To wit, the outflow at the two weirs into canal–2 and canal–1 is based on Bernoulli's relation and flow criticality, cf. Munson et al. (2005). At $s = L_{1c}$, e.g., for subcritical flow with flow depth $h_{2c}$ and flow speeds $V_{2c} \approx 0$ upstream as well as critical flow $V_c$ of height $h_c$ over the weir, we therefore derive the following

$$V_c = \sqrt{gh_c} \quad \text{and} \quad gh_{2c} + \frac{1}{2}V_{2c}^2 = g(h_c + P_{2w}) + \frac{1}{2}V_c^2 = \frac{3}{2}gh_c + gP_{2w}$$

$$V_{2c} \approx 0 \quad \text{s.t.} \quad h_c = (2/3)(h_{2c} - P_{2w}) \quad \text{and therefore:}$$

$$20 \quad Q_{2c} = w_c h_c V_c = w_c \sqrt{g} h_c^{3/2} = C_f \sqrt{g}\, w_c \max(h_{2c} - P_{2w}, 0)^{3/2} \tag{13}$$

with $C_f = (2/3)^{3/2}$. Similar derivations with suitable adaptations of the quantities involved determine the fluxes $Q_{1c}, Q_{3c}, Q_{res}$ over the other weirs.

### 2.2.5 Fully-coupled system

When all of the above models for the individual components are combined we obtain the entire coupled model, including its initial and boundary conditions, for the unknowns $h(s,t)$, $h_m(y,t)$, $h_{res}(t)$, $h_{1c}(t)$, $h_{2c}(t)$ and $h_{3c}(t)$:

$$\text{River:} \quad \partial_t(w_r h) + \partial_s\left(w_r h R(h)^{2/3}\sqrt{-\partial_s b}/C_m\right) = Q_m\delta(s-s_m) + Q_{res}\delta(s-s_{res}) + Q_{1c}\delta(s-L_{1c})$$

$$\text{on} \quad s \in [0,L] \quad \text{with} \quad Q_f|_{s=0} = w_r h R(h)^{2/3}\sqrt{-\partial_s b}/C_m|_{s=0} = Q_0(t), \quad h(s,0) = h_0(s) \tag{14a}$$

$$\text{Moor:} \quad \partial_t(w_v h_m) - \alpha g \partial_y\left(w_v h_m \partial_y h_m\right) = \frac{w_v R_m(t)}{m_{por}\sigma_e}$$

$$\text{on} \quad y \in [0,L_y] \quad \text{with} \quad \partial_y h_m|_{y=L_y} = 0, \quad h_m(0,t) = h_{3c}(t), \quad h_m(y,0) = h_{m0}(y) \tag{14b}$$

$$\text{Reservoir:} \quad w_{res}L_{res}\frac{\mathrm{d}h_{res}}{\mathrm{d}t} = w_{res}L_{res}R_{res}(t) - Q_{res} \quad \text{with} \quad h_{res}(0) = h_{r0} \tag{14c}$$

$$\text{Canal-1:} \quad w_c(L_{1c}-L_{2c})\frac{\mathrm{d}h_{1c}}{\mathrm{d}t} = Q_{2c} - Q_{1c} \quad \text{with} \quad h_{1c}(0) = h_{10} \tag{14d}$$

$$\text{Canal-2:} \quad w_c(L_{2c}-L_{3c})\frac{\mathrm{d}h_{2c}}{\mathrm{d}t} = Q_{3c} - Q_{2c} \quad \text{with} \quad h_{2c}(0) = h_{20} \tag{14e}$$

$$\text{Canal-3:} \quad w_c L_{3c}\frac{\mathrm{d}h_{3c}}{\mathrm{d}t} = \gamma Q_{tm} - Q_{3c} \quad \text{with} \quad h_{3c}(0) = h_{30}, \tag{14f}$$

with

$$Q_{1c} = C_f\sqrt{g}\,w_c\max(h_{1c}-P_{1w},0)^{3/2} \quad \text{and} \quad Q_{2c} = C_f\sqrt{g}\,w_c\max(h_{2c}-P_{2w},0)^{3/2} \tag{14g}$$

$$Q_{3c} = C_f\sqrt{g}\,w_c\max(h_{3c}-P_{3w},0)^{3/2} \quad \text{and} \quad Q_m = (1-\gamma)Q_{tm} \equiv (1-\gamma)\frac{1}{2}m_{por}\sigma_e W_v\alpha g(\partial_y h_m)^2|_{y=0} \tag{14h}$$

$$Q_{res} = C_f\sqrt{g}\,w_{res}\max(h_{res}-P_{wr},0)^{3/2} \quad \text{and} \quad R(h) = w_r h/(2h+w_r), \tag{14i}$$

as well as time-dependent rainfall functions $R_{res}(t), R_m(t)$ and upstream inflow $Q_0(t)$. The remaining parameters are constants, with units and typical values listed in Table 2. The rainfall functions are defined such that, in the absence of other effects, e.g. unit porosity in the moor, they directly lead to a linear increase of the moor's ground water level and the reservoir depth. Note that the mass fluxes involving the different $Q$'s between the various systems are consistent. A space-time numerical discretisation of (14) is given in Appendix A. It involves a second-order finite-difference approximation of the ground water equation (14b) in $y$, a first-order finite-volume discretisation of the river equation (14a) in $s$, and straightforward first-order forward-Euler time discretizations of the time derivatives involved.

The rainfall functions are constant during on every wd and generally vary from Wetropolis day-to-day. On a given Wetropolis day,

$$R_{res}(t) = n_r n_{res} r_0 \quad \text{and} \quad R_m(t) = n_r n_{moor} r_0, \tag{15}$$

in which $n_r = 1,2,4,9$ is drawn daily with probability $(3,7,5,1)/16$ via one Galton board, while one of the combinations

$$(n_{res},n_{moor}) = \{(1,0),(1,1),(0,1),(0,0)\}$$

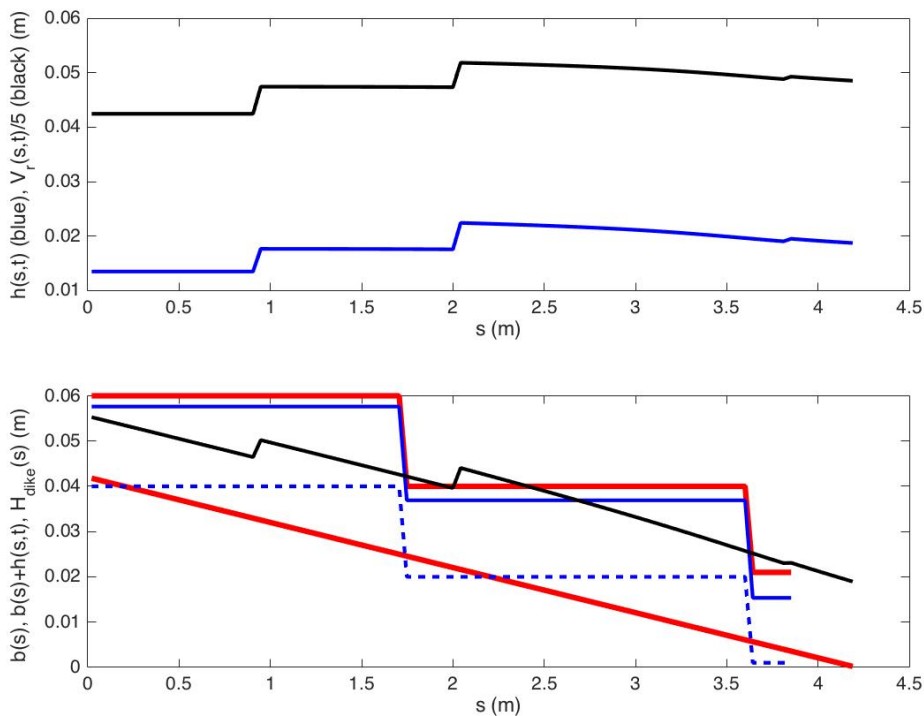

**Figure 4.** Top panel: the river level $h(s,t)$ of the river (blue, bottom) and the river velocity $V_r(s,t)$ (black, top) as function of the along-river coordinate $s$ at $t = 5000$s; the bottom panel: topography $b = b(s)$ (in red and fixed), the top of the berm or dike along river/canals in red (fixed); in dashed blue the bottom of the set of canals; in solid blue the canal levels; in black is the dynamic river level indicated above the bed; all as function of $s$ at time $t = 5000$s. When the black line/river level lies above the red lines/berms there is flooding, here because at $t = 5000$s the water level is seen to be high, cf. Fig.5 bottom-right panel. The black line is seen to have three jumps at $s = s_{res} = 0.925$m, $s = s_m = 2.038$m and a small one at $s = L_{1c} = 3.858$m where water comes in from the reservoir, moor and canal respectively. At $s = 0$ there is constant water influx. Flooding is just defined as water-level exceedance above the canal berm: in this simplified design model there is no actual water leaving the river.

is drawn daily with probabilities $(3, 7, 5, 1)/16$ via the other Galton board, as explained in §2.1. The rainfall speed $r_0$ will be determined by trial-and-error and has the units of $\partial_t h$, i.e. m/s. Hence, the volumetric rate of rainfall per wd on the moor for unit $n_r = 1$ can then be calculated, yielding

$$V_{rate} = (L_y w_v r_0)\,\text{wd}. \tag{16}$$

**Table 2.** Parameters: units and values used. Note that $\alpha = k/(m_{por}\nu\sigma_e)$. $\gamma \in [0,1]$.

| Parameter | Units | Value | Parameter | Units | Value |
|---|---|---|---|---|---|
| $g$ | m/s$^2$ | 9.81 | $P_{1w}$ | m | 0.01 |
| $L$ | m | 4.211 | $P_{2w}$ | m | 0.0125 |
| $C_f$ | - | $(2/3)^{3/2}$ | $P_{3w}$ | m | 0.0125 |
| $C_m$ | m$^{-1/3}$s | 0.02 | $L_{1c}$ | m | 3.858 |
| $\mathrm{d}b/\mathrm{d}s$ | - | 0.01 | $L_{2c}$ | m | 3.608 |
| $w_r$ | m | 0.05 | $L_{3c}$ | m | 1.724 |
| $w_v$ | m | 0.095 | $s_{res}$ | m | 0.932 |
| $L_y$ | m | 0.925 | $w_{res}$ | m | 0.123 |
| $m_{por}$ | - | 0.3 | $L_{res}$ | m | 0.293 |
| $\sigma_e$ | - | 0.8 | $P_{wr}$ | m | 0.1 |
| $k$ | $m^2$ | $10^{-8}$ | $s_m$ | m | 2.038 |
| $\nu$ | m$^2$/s | $10^{-6}$ | $\gamma$ | - | 0.2 |
| $w_c$ | m | 0.02 | | | |

## 2.3 Numerical results

Given the choice of parameter values with (or near) the values given in Table 2, the goal is to determine a suitable rainfall speed $r_0$ and length wd via trial-and-error through numerical simulation. As initial conditions we take $h(s,0) = 0.0135$m, $h_m(y,0) = 0$, zero canal and reservoir levels $h_{10} = h_{20} = h_{30} = h_{res0} = 0$, and an upstream influx of river water corresponding to the mass flux $Q_0 = Q_f(h(0,0))$ associated with $h(0,0)$. In reality, rainfall will be varied daily by changing the action of two pumps, which require a fraction of a second to change gear. For someone viewing Wetropolis, a length of day between 5s and 20s seems reasonable so we have chosen wd $= 10$s as a first guess. In the design phase, values of $r_0$ have been chosen and tuned in simulations of 100 to 500wd's, i.e. 1000s to 5000s, which can be simulated in about $10\%$ of real time. Hence, with the choice wd $= 10$s, the *return period of extreme rainfall in Wetropolis* is therefore $(256/7)\,10\mathrm{s} = 6:06$min; and, the chance of two consecutive days of extreme rainfall has a return period of $(256/7)^2\,10\mathrm{s} = 223$min $\approx 3:43$hr.

To monitor whether $r_0$ has the (approximately) desired value during a simulation, major flooding is defined to occur when the river level significantly, i.e. by circa $0.01$m or more, exceeds the canal–1 berm along the strip of river bordering the city plain. This is monitored visually in daily snapshots, one of which is given in the lower panel of Fig. 4. It contains a compound of levels to enable this flood monitoring, which requires some explanation. While the canal water enters the river in the city, for simplicity flood waters of the river are not modelled numerically to enter the canal or city, which suffices for our design purposes. The information displayed in the lower panel of Fig. 4 is as follows. The vertical axis has units in m so the range across the length $L = 4.21$m of the set-up is about $0.06$m with the horizontal $s$–axis lying along the river and canal. The zero-level of the canals and rivers in the vertical is put at the river exit $(s,z) = (L,0)$ with vertical coordinate $z$. In reality, the lengths of river and canal differ slightly owing to the curved channels; for design and our numerical model, this difference is considered negligible and it is sufficient to assume they are of the same length. The lower, solid and thick red line displays the

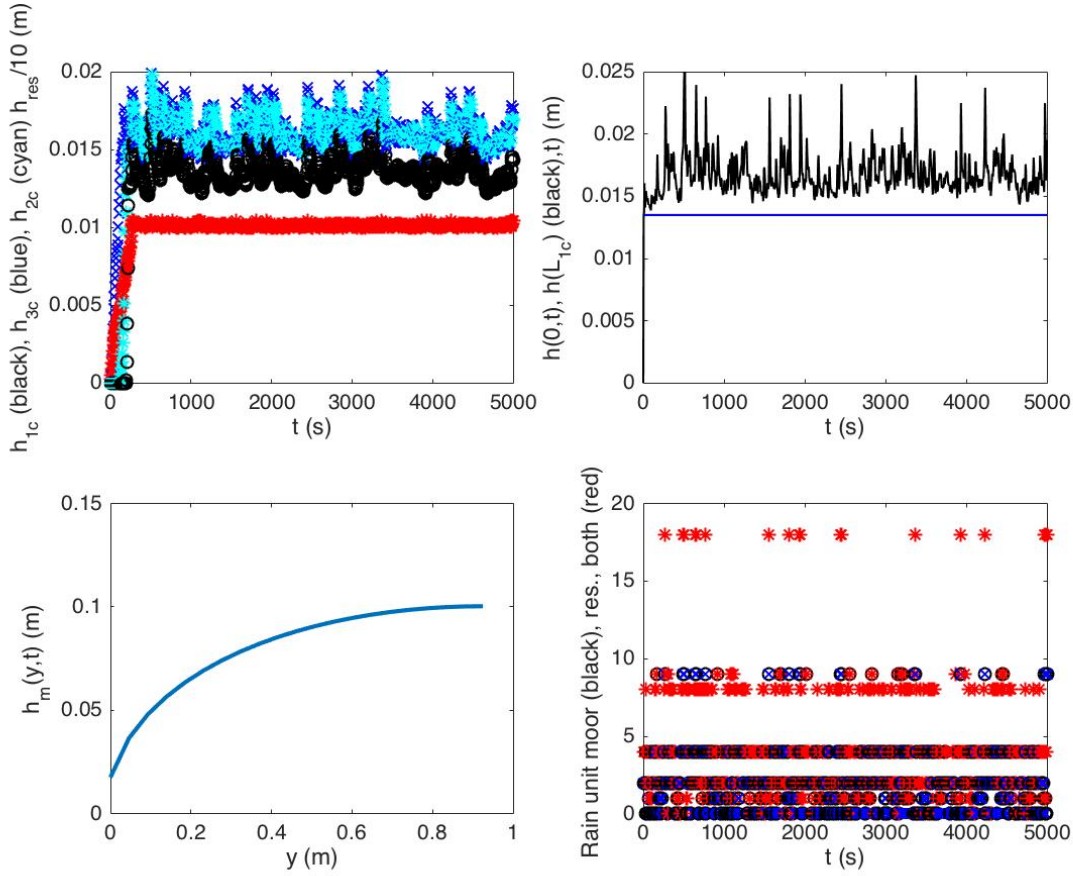

**Figure 5.** A four-panel figure in which: (i) the first top-left panel contains the three canal levels and the level of the reservoir as function of time $t$ all initialised at zero in this run (reservoir: red; canal–1: black, canal–2: blue; canal–3: cyan); (ii) the second top-right panel displays the river level at $s = 0$ in blue and the river level at one point in the city in black as function of time; (iii) the bottom left panel shows moor groundwater level $h_m(y,t)$ as function of space $y$ in a snapshot at $t = 5000s$; and, (iv) panel four, bottom right, shows rainfall per $wd = 10s$ scaled with the magic factor $r_0$ versus time.

fixed sloped bottom of the river (with 1% downhill gradient, i.e., $\partial_s b = -0.01$). The thinner solid-black line displays the river level with the upstream input depth of $h(0,t) = h_0(t)$. This water profile is generally adjusting dynamically to become uniform except at the reservoir influx ($s_{res} = 0.932$m), the moor influx ($s_m = 2.038$m) and the canal–1 influx ($s = 3.858$m), which cause sudden increases in the river levels. These larger and smaller jumps are indeed visible and identifiable in the lower panel of Fig. 4. The flux into the river from canal–1 is comparatively small, so the rise in the river level here is much smaller than the time-varying influx of water from the reservoir and moor. The bottom of the three canal sections is displayed by the (stepped) thick-blue dashed line with the upper canal–3 level at $z = b_3 = 0.04$m, the middle canal–2 level at $z = b_2 = 0.02$m and the lower canal–1 level at $z = b_1 = 0.0$m. The three berm (or dike) heights are 0.02m higher at $\{d_3, d_2, d_1\} = \{0.0, 0.04, 0.02\}$m.

Canal berm or dike levels are displayed with a (stepped) thick solid-red line, while the three varying canal levels are displayed as the (stepped) solid-blue line. Steps in the berms occur where the weirs are placed and the jumps in the varying canal levels are determined by the hydraulic weir relations at these weirs. Some river flooding can occur when the river level, the (stepped) solid-black line, exceeds the canal–2 berm downstream of the second weir, as is visible in the lower panel of Fig. 4. *Major flooding is defined when the river levels exceeds the canal–1 berm in the city section, i.e. at $s = L_{1c} = 3.858$m the water depth $h(L_{1c}, t)$ significantly exceeds $0.02$m*, visible as snapshot in the lower panel of Fig. 4, where the solid-black line of the river level is seen to exceed the solid-red canal–1 berm level downstream of the last weir around $s = 3.7$m. Via visual optimisation, i.e., monitoring when major flooding occurred in the city for the extreme or rare events of $90\%$ rainfall in both the reservoir and moor, a suitable value of the rainfall speed is found to lie around the value

$$r_0 = 2.05 10^{-4} \text{m/s}. \tag{17}$$

The corresponding water volumes for the various Galton-board outputs required in the moor per wd are then

$$(1, 2, 4, 8, 9, 18) V_{rate} = (0.18, 0.36, 0.72, 1.44, 1.62, 3.24) \text{l/wd} = (0.018, 0.036, 0.072, 0.144, 0.162, 0.324) \text{l/s}. \tag{18}$$

Consequently, the pump supplying the rainfall on the moor should have a maximum discharge of about $324$ml$/s$, which is a manageable amount from a design perspective. Such a discharge is feasible by using inexpensive off-the-shelf aquarium pumps, both for the supply of the upstream river influx $Q_0$ and the varying rainfall amounts in reservoir and moor.

An example of a simulation over $500$wd's is summarised in Fig. 5. Since reservoir and canals are empty at $t = 0$, we observe that it takes about $25$wd's before they are filled. During this time major flooding is lessened, or prevented completely, because the reservoir and canal in essence act like flood-attenuation storage sites, supplying passive flood control. Extreme rainfall in this start-up period tends to be buffered such that city flooding is prevented. Reservoir and canal levels are displayed in the top-left panel of Fig. 5 versus time. The (constant) upstream river level $h(0, t)$ and city river level $h(L_{1c}, t)$ are displayed as function of time $t$ in the top-right panel of Fig. 5, in which extreme events with $n_r = 18$ are clearly identifiable as flood peaks at time $t$ when $h(L_{1c}, t) > 0.02$m. A snapshot of the groundwater level $h_m(y, t)$ in the moor is displayed in the bottom-left panel of Fig. 5; it shows the no-flux upstream boundary condition at $y = L_y$ and the gradual decrease of the groundwater level towards its outflow location at $y = 0$. The rain units for the moor (being $1, 2, 4$ and $9$), reservoir (being $1, 2, 4$ or $9$) and their summation $n_r$, with the discrete values of $1, 2, 4, 8, 9$ or $18$, are displayed in the bottom-right panel of Fig. 5. The peaks of extreme rainfall are, of course, seen to match the peaks in extreme flooding in the panels on the right except, possibly, during the first circa $25$wd's when the canals and reservoir tend to act as flood-attenuation buffers.

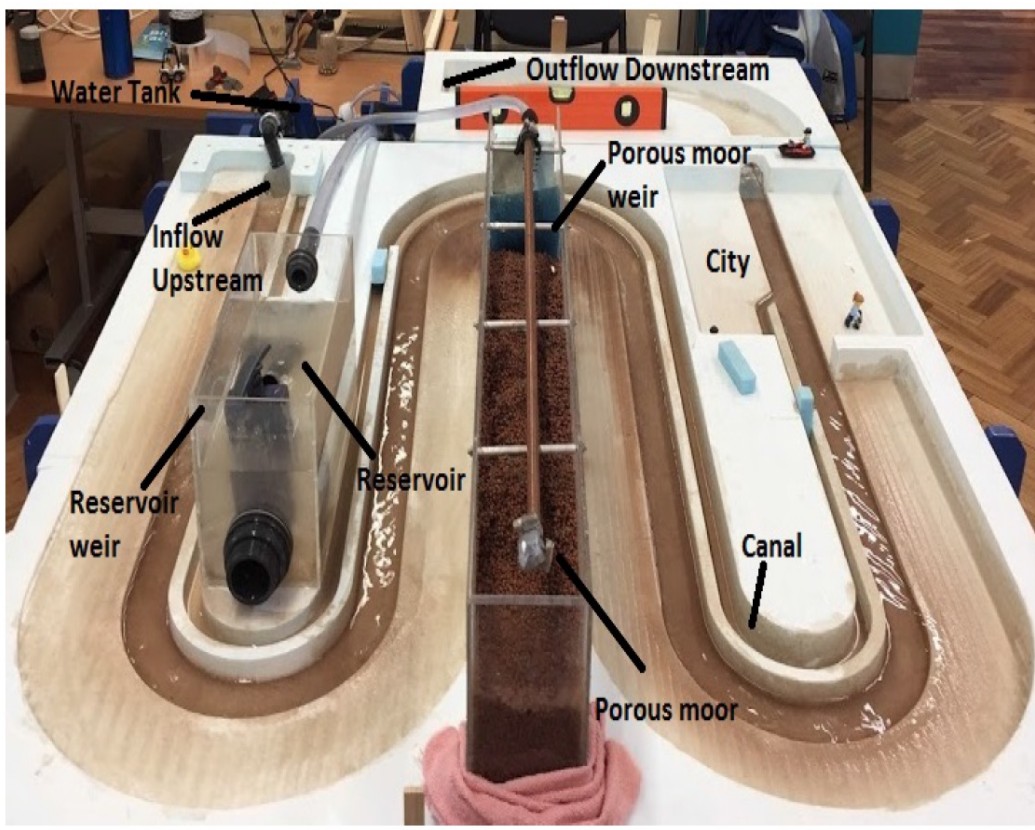

**Figure 6.** Overview of the Wetropolis flood demonstrator with its winding river channel of circa 5.2m and the slanted flood plains on one side of the river, a reservoir, the porous moor, the (constant) upstream inflow of water, the canal with weirs (the three small blue foam-wedges seen in the photograph), the higher city plain, and the outflow in the water tank/bucket with its three pumps. Two of these pumps switch on randomly for $(1, 2, 4)$ or 9s of each wd $= 10$s. Photo compilation: Luke Barber.

## 3 Table-top design

After the design calculations commenced on May $29^{\text{th}}$ 2016 and were completed on June $8^{\text{th}}$ 2016[3], the Wetropolis flood demonstrator was constructed and finalised between June $4^{\text{th}}$ and August $31^{\text{st}}$ by OB and WZ[4]. The final design was limited by the demand to transport it in the back of a car. We note that the reservoir and moor have been swapped in the actual set-up, relative to the mathematical design, and that the river-channel length has been increased to 5.2m.

The physical Wetropolis model consists of several elements which we describe next:

---

[3]The first design and complete model calculations were presented during a seminar at Imperial College London on June $1^{\text{st}}$ 2016. A week later an error in the use of the Manning coefficient $C_m$ was fixed, leading to an increase of the river channel length $L$ by a factor of four. Hence, the winding channel.

[4]See public postings in that period around 08-06-2016 and 31-08-2016 on https://www.facebook.com/resurging.flows and the design history on https://github.com/obokhove/wetropolis20162020 .

- The topographic landscape with a winding river channel, one-sided flood plain, canals and the city-plain has been routed out of two standard polystyrene foam plates each of dimension $5 \times 60 \times 120 \text{cm}^3$ (plus a small extra foam plate) with an overlay to fit two plates together. A smaller third piece was added to extend the river length after the city which enhanced flooding in the city plain. An overview is given in Fig. 2 and a photograph is found in Fig. 6. Drawings have been made in the CAD programs Rhino/Grasshopper and used to steer the router. Routing precision is circa $0.8\text{mm}$. After the routing, the river channel and its flood plain have been roughened by varnishing fine sand to the base.

- A framework of wooden support slabs has been made that fits on four A-frames. This framework is put together with a bolt-nut system such that it can be disassembled for transport. Wooden wedges are used to squeeze and level the foam pieces within the slab-framework. The three foam pieces are squeezed together to limit leakage. Aluminium one-side-sticky tape is used locally to seal two sections of the river channel together and thus bridge two adjacent channels. Rather than sealing off all leakage, which in practice becomes impossible, an "aquifer" system of two interconnected gutters underneath the seams of the foam pieces leads leaked water back to the holding reservoir with the three aquarium pumps. Hence, the water budget is closed in the absence of evaporation, the latter which is negligible on the time scale of operation (typically a few hours) but not on the time scale of a day or more.

- Three aquarium pumps with a maximum pumping capacity of $0.375\text{ml/s}$ are placed in a holding reservoir, a rectangular bucket with dimensions $\sim 0.3 \times 0.40 \times 0.22 \text{m}^3$, which bucket or reservoir is hung underneath the wooden framework in a rectangular area adjacent to the upstream inflow point of river water. Plastic tubing with inner and outer diameters of circa $(1.8, 2.2)\text{mm}$ leads the water from this reservoir to the upstream point and, depending on whether it rains or not, to the reservoir and the moor. Rainfall on the moor is spread out and visualised using a copper pipe with numerous downward-facing holes over the lava grit.

- The moor unit is made of acrylic and on the open side-face of the box a gauze prevents the lava grit from avalanching into the river. The acrylic reservoir is open from the top and water can enter through a hole near the top edge. Outflow of water in the river is regulated via an internal pipe which outflow level can be manually adjusted. Hence, active flood control can be demonstrated by manually adjusting this outflow level. Outflow into the canal can be arranged separately via an adjustable valve. Note that this is slightly different from the set-up in the mathematical and numerical design model, where the outflow of moor water was partitioned between the river and canal.

- The set-up for the Galton boards including the Galton boards themselves, the accompanying Arduino control units, power sockets and plugs to operate the three aquarium pumps. The two draws from the discrete probability distribution are either computer generated or determined from the random paths of (a) steel ball(s) through the asymmetric Galton boards. In the latter case, the steel ball triggers a signal by interrupting optical sensors in one of the four channels on each Galton board, cf. Fig. 1. The signal subsequently steers either the reservoir pump, moor pump, both or none as arranged via the Arduino technology.

Further specifications have been provided in Appendix B.

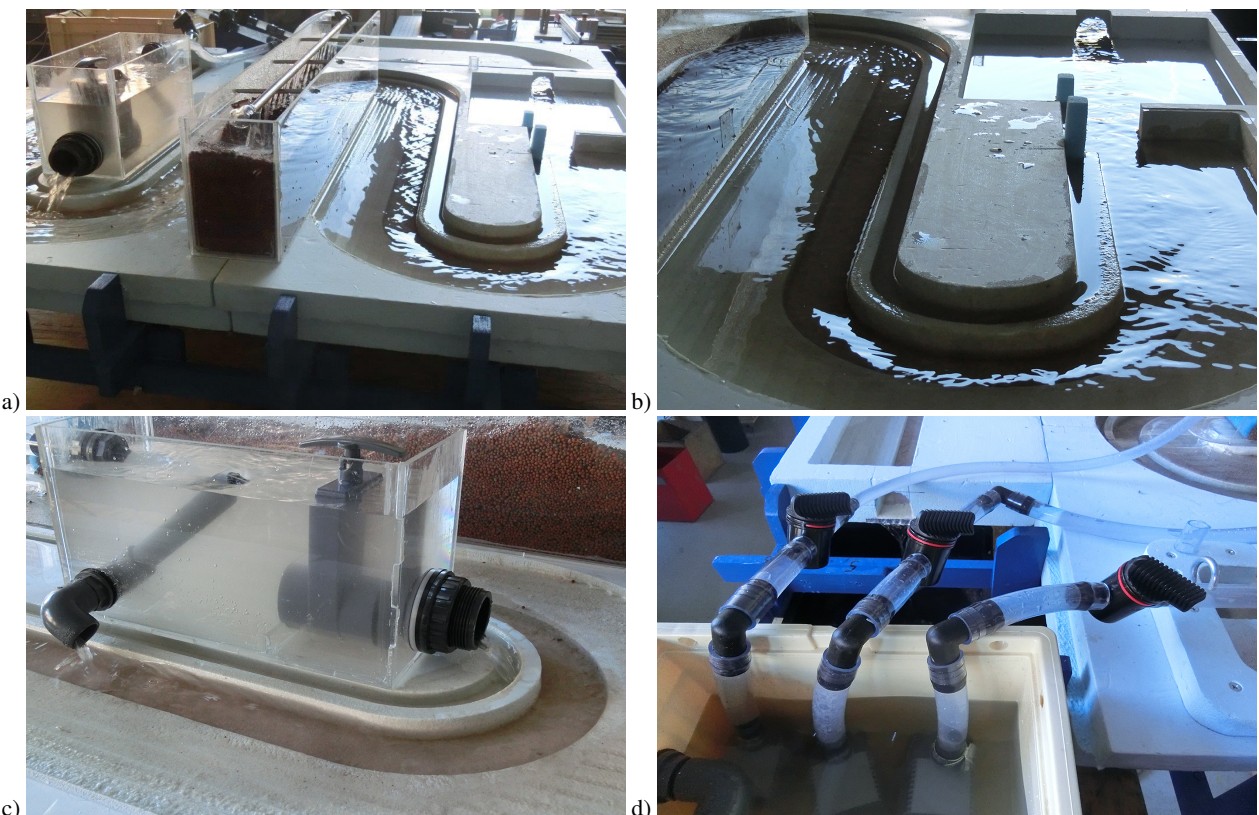

**Figure 7.** Action shots of Wetropolis: a) overview of overflowing reservoir on the left, the lave-grit filled moor under heavy rainfall in the middle and a flooded city in the background on the right during tests with massive flooding and $100\%$ rainfall over several days; b) zoom-in of the final river bend and its one-sided flood plain and the canal before the city as well as a flooded city plain in the background on the right during massive flooding; c) zoom-in of the reservoir with water streaming through the manually adjustable outflow pipe into the river and the separate valve-adjustable underflow into the canal on the right; and, d) zoom-in of the holding reservoir with the three aquarium pumps and tubing leading to the constant upstream inflow at the start of the river at $s = 0$ on the right and two other tubes leading to the reservoir and moor.

## 4 Wetropolis illustrated and demonstrated

Illustrative images of Wetropolis in action are shown in the photographs of Figs. 7 and 8. It includes close-ups of excessively flooded river-bends and city plain, the reservoir and its outlets as well as the moor under heavy rainfall in Fig. 7. During extremely heavy rainfall (90% per $\mathrm{wd}$) after a relatively wet period, the moor becomes supersaturated and the groundwater level can rise through the lava grit and trigger fast surface run-off. In other situations the groundwater level is below the surface of the lava grit. Under varying rainfall the rising and falling groundwater level can be observed through the transparent acrylic walls. This visualisation was inspired by cartoons in hydrological textbooks, in which we often find similar cross-sections of the earth and its groundwater levels.

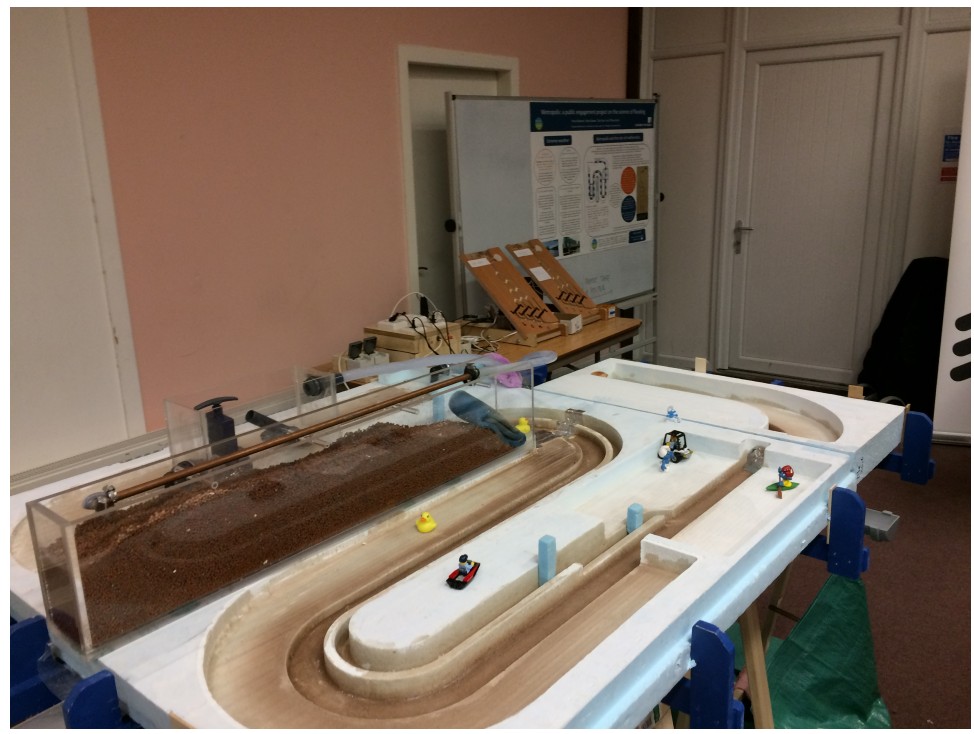

**Figure 8.** Photograph of the entire set-up at the Churchtown Flood Action Group workshop on 28-01-2017, with the winding river channel in the foreground, the city plain with a few smurfs, the groundwater moor, the reservoir on the left behind the moor, and in the background on the right, the control table with the Arduino units and the two Galton boards as well as an informative poster on Wetropolis.

To date Wetropolis Flood Demonstrator has been showcased to flood victims at public events, attendees of science fairs, and to scientists and flood professionals at bespoke workshops on natural hazards. Wetropolis has been showcased [5]:

– first at the general assembly of the EPSRC UK network Maths "Foresees" in Edinburgh in September 2016 to an audience of scientists with expertise in environmental fluid dynamics and representatives from stakeholders such as the Met Office, JBA Trust and the Environment Agency; Wetropolis has been created as an outreach project within this Maths Foresees network;

– at the Churchtown flood action group conference in January 2017 in Lancashire to circa 140 flood victims and several experts on flooding, after a public lecture by OB on the statistics of extremes, see Fig. 8;

– two afternoons as part of the public exhibition on the Boxing Day 2015 floods in Leeds' Armley Industrial Museum in December 2017 and March 2017 to an audience of flood victims, family and friends, including children;

---

[5]For movie footage see the posts dated 31-08-2016 [with an extremely rare Boxing-Day-2015-type flood after two consecutive days of extreme rainfall], 06-09-2016, 16-01-2017, 08-12-2016, and 07-04-2017 on https://www.facebook.com/resurging.flows, or at https://youtu.be/1FIHFOn6IPQ and the Boxing-Day-equivalent flood at https://youtu.be/N4Sp5gHXcz4

- at a bespoke Canal & River Trust workshop in Liverpool in March 2017 by students of Leeds Centre for Doctoral Training in Fluid Dynamics;

- at the "Be Curious" science festival in March 2017 at the University of Leeds for a general audience;

- to scientists at the Maths Foresees "Environmental Modelling in Industry" study group at the Newton Gateway to Mathematics, Cambridge, in April 2017, which particular event focussed on solving mathematical challenges related to flooding, see https://gateway.newton.ac.uk/event/tgmw41; and,

- at the Yorkshire iCASP confluence (integrated catchment program) in June 2018 for a range of scientists, flood professionals, stakeholders and politicians, see the right panel of Fig, 1.

Based on interactions during the above events with audience members and organisers, as well as on formal discussion during workshops regarding public engagement, we consider some of Wetropolis' strengths and weaknesses. Most of these considerations are anecdotal except for the discussions at "Maths Foresees" meetings and the Newton Gateway to Mathematics, which were based on formal notes of the in-depth round table and workshop discussions as well as the study-group host. We stress, therefore, that the outreach component of this study is lacking a proper scientific method (from a social-science perspective). Indeed, none of these discussions concern formal questionnaires, well-balanced questions and subsequent statistical analysis, which would constitute a formal investigation of the feedback and Wetropolis' impact. There are two reasons why such a formal analysis is lacking: the authors do not have the expertise to undertake such an analysis and, more importantly, during the showcasings for at-the-time recent flood victims we did not want to further bother these victims by conducting intrusive questionnaires in a potentially unhelpful manner. However, this is something to be considered for future demonstrations, in collaboration with the necessary experts. It also implies that the conclusions suggested below are preliminary in nature.

The strength of Wetropolis is that it is a physical visualisation of the probability of extreme rainfall and flooding events including actual and visual river hydraulics, groundwater level changes and interactive flow control. We recall that the reservoir has valves such that the audience can store and release water interactively into the canal and river in order to control and possibly prevent flooding in the city. Wetropolis is, however, a conceptual model of flooding rather than a literal scale model of a specific catchment. It has, however, been inspired by the Boxing Day Floods of 2015 of the River Aire, in and upstream of Leeds, UK. This conceptuality is both a strength and weakness because one needs to explain the translation of a $1 : 200$ year return period for a realistic extreme flooding and rainfall event such as the Boxing Day 2015 flood of the River Aire into one in Wetropolis with its one in $6{:}06\mathrm{min}$ return period, and one also needs to explain that the moor and reservoir are conceptual valleys where all the rain falls, since rain cannot fall everywhere in the Wetropolis catchment, in contrast to rainfall in real catchments. This scaling and translation step is part of the conceptualisation, which the audience, whether public or scientific, needs to grasp. The visualisations of flooding in the city and the ground water level also involve learning steps. Hitherto, this conceptualisation step was either explained by the Wetropolis wardens in attendance at a demonstration, via our bespoke poster, or both. Alternatively, we aim to arrange bespoke audiovisual material.

Due to this learning curve, the most receptive public audiences have been flood victims or people with friends or family who went through the unpleasant and potentially devastating experience of being flooded (cf. two showcasings at Armley Musuem

flood exhibition and Churchtown Flood Action group workshop). We have perceived such audiences to be the most receptive, inquisitive and interactive because they have an intrinsic interest in flooding phenomena and wish particular questions to be addressed in order to gain more understanding as to what causes flood hazards, how these hazards can be predicted and how such floods can possibly be tackled through flood mitigation and/or management. Recall that these were exactly the questions with which we started off in the introduction. In particular, Wetropolis aids in raising awareness of the probabilistic character and randomness of rainfall and flooding events, also in connection with the difficulties in predicting some of these extreme events. Combining showcasing Wetropolis with a general public lecture on the science of flooding has proven to be particularly successful, cf. (Science of floods, 2016; Potter, 2016), owing to such a presentation whetting the appetite to view a scale model with rainfall and river flooding. While Wetropolis was designed as a public outreach project, the reception from flood practitioners and scientists working in environmental fluid dynamics has been surprisingly positive.

Finally, we recently showcased Wetropolis II (a new and improved demonstrator based on the same design principles presented here) as part of the (biannual) Mathematics of Planet Earth exhibition[6] organised by the corresponding Centre for Doctoral Training at Imperial College in London for about 500 to 1000 visitors over nine days, from February $15^{th}$ to $23^{rd}$ 2020. In line with the directives of the organisers, the audience was encouraged to volunteer bespoke feedback on two post-it boards, one for the larger exhibition and one specifically for Wetropolis. We received ten feedback posts with positive feedback as well as suggestions, the latter ranging from: (a) build more of these, (b) make an exhibition set-up for tsunamis to (c) please add a full-fledged rain and river flow predictive model of Wetropolis and compare the two[7]. While this feedback is still lacking an in-depth statistical basis, given that the organisers felt that a formal questionnaire would be too intrusive, it does provide additional insights into the audience' perception of Wetropolis.

## 5  Summary and discussion

We have demonstrated how the Wetropolis demonstrator of extreme rainfall and floods was constructed after an efficient mathematical and numerical design enabled us to estimate the characteristic components of the envisioned set-up. This efficient mathematical model was first presented as a coupled system of ordinary and partial differential equations which we subsequently solved numerically to define a near-optimal design. While that mathematical model is close to a prediction model for river and groundwater flows in Wetropolis, due to its relatively minimalist nature and purpose to facilitate only the design, it is likely not quite sophisticated enough to make bonafide predictions. In a final modelling step, we determined the reasonable rainfall and flow rates through numerical simulation, on which rates we based the actual design and construction of Wetropolis. We highlight that we have provided all these design steps, and made all software and design drawings fully available, in order to facilitate design adaptations by the reader, to conceptualise their particular catchments of interest in their own Wetropolis. Several improvements and extensions of Wetropolis are under exploration, as follows:

---

[6]See: http://www.imperial.ac.uk/news/195539/planet-earth-alive-using-mathematics-understand/ .

[7]This feedback and the feedback of all exhibitions listed above are found at https://github.com/obokhove/wetropolis20162020/tree/master/feedback

- to accentuate a flooding event in the city more prominently, e.g. by measuring the flood waters of each flood event via a separate drain into a measuring cup or by triggering flashlights in the city to light up by closing an electric circuit by the flood water and to go off when the circuit is broken; in Wetropolis II, we have added a drain in one half of the city, with its land lying somewhat lower and protected by a dike, such that the volume of each flood is visually captured in a beaker;

- to visualise key principles of Natural Flood Management (NFM) or more broadly Nature Based Solutions (NBS) (e.g., Hankin et al. (2017); Lane (2017); Potter (2016); Cabaneros et al. (2018); Bokhove et al. (2018b, 2019)) by visualising the effects of different riverbed roughness to slow down the flow in a cut-out river-bed segment via removable river-channel inserts and by including a porous upland catchment with various small-scale river channels and flow-attenuation features to enhance water storage, the features of which can be manipulated by the audience (operational in Wetropolis II);

- to include droughts by modification of the dry days visualised by drying up of a water supply pipe line from the moor to the city, e.g., given the extremely dry summers of 1976 and 2018 in Europe; and,

- to include climate change by making certain flood events more extreme (i.e., by varying the discrete probability distribution); climate change is included in Wetropolis II by the addition of an extra upstream reservoir and corresponding pump, which unit is synchronised with the moor operation through the Galton boards, while the primary reservoir has been moved downstream near the city plains; via a switch this extra water influx due to "climate change" can be switched on or off.

Droughts can be modelled by extending $1:4$ of the dry days/periods, i.e., with their $1/16$ chance, to four or five dry days, while the other three outcomes are kept at one dry day, with suitable changes of the overall statistics. Since the Galton board outcome on rainfall amount is not used on dry days, its outcome can be recycled to randomly assign for the drought period as $1:4$. Climate change can be modelled by extending extreme rainfall to two or three days for on average $1:3$ extreme rainfall outcomes, or per another ratio, in a similar fashion. These drought or climate change adaptations can be included into the same physical set-up via changes in the Arduino programming.

## 5.1 Games

One of the shortcomings of the current Wetropolis set-up is that it lacks bespoke educational material and games. The game suggestions for the current Wetropolis set-up, which have arisen in evaluation sessions during some of the various workshops mentioned above, are as follows:

- The notion of (theoretical and sampled) pdfs can be developed, based on recording histograms of actual Galton board outcomes and comparison of these outcomes against the theoretical outcomes. Games can be created to determine whether the outcomes have been tampered with by human intervention, e.g., a game in which one team is allowed to trigger extreme flooding through tricking the electronic eyes by a finger or by purposely misaligning the Galton boards without

telling the other team whether or not such unnatural interventions took place. While the audience generally likes massive flooding to occur more often in Wetropolis, by secretly triggering daily $90\%$ rainfall in both moor and reservoir, recording the Galton board outcomes would immediately reveal that such tampering is unrealistic in that it makes no rain and low to intermediate rainfall into rare rather than common events.

– Building a game on flood prevention in the city by controlling the valves on the reservoir, say over 30 to 100wd's, with the winning team having the least or zero amount of flooding in the city. This can also include a discussion on the possibility that the winning team can win by chance over a limited set of trials rather than by virtue of optimal flood control.

– The audience can play with the set-up. The set-up can namely be modified by interchanging the two-out-of-three locations where rainfall is random, changing the locations of the reservoir and moor, for example by bringing one unit in close proximity to the city, including an investigation as to what consequences these changes entail in observed spatio-temporal rainfall and flooding patterns.

– A particular case can be chosen through changing a switch between normal, drought and climate settings; and, a game can be created for teams, to determine which probability distribution is used from the observed outcomes. Given the inevitable bias in the analogue Galton boards and the nature of statistics, such a game will include some uncertainty. Finally, when the Galton boards fail, which happens occasionally if a steel ball balances exactly on a split, then an automatic routine takes over. Calculating the changes in the return periods, for the droughts, extreme events and super-extreme climate-change events is an interesting yet straightforward (class-room) exercise.

– Recently, it has been suggested to make a full predictive numerical model and include a few measurements with data assimilation in order to facilitate a live scientific display of the weather and river flow predictive capabilities.

– Finally, a formal statistical evaluation of the response of the public to Wetropolis has not yet been undertaken.

Each of these suggestions requires further development.

## 5.2 New approaches in science and water management

Flood practitioners from various stakeholders have been quite positive about Wetropolis' novel way to visualise the probability of extreme rainfall and flooding events via the asymmetric Galton boards and how the outcomes from these boards directly lead to observable rainfall and river dynamics in the set-up. Stakeholders such as JBA Trust and the Environment Agency see Wetropolis as a potentially useful tool to trigger discussions about innovations in flood mitigation and water management, as part of workshops and brainstorm sessions. To date, Wetropolis has triggered two innovations: one on the use of the revisited concept of flood-excess volume (FEV) in devising a novel and graphical cost-effectiveness analysis to flood mitigation, in particular meant for decision makers, and one on education in water engineering and management.

Flood-excess volume (FEV) concerns the volume of flood waters that caused flood damage. It is the flood volume of the river flow beyond a certain, chosen and relevant threshold water level at a certain target, critical river location. This FEV, expressed in

cubic metres ($\mathrm{m}^3$), or expressed more visually and comprehensively as a square lake of 2m in depth with a certain side length, is a useful measure to devise flood-mitigation strategies. It allows us to quantify what fraction of the FEV is mitigated by a certain strategy. Our cost-effectiveness analysis Bokhove et al. (2018a, 2019) results in a series of square-lake graphs, one for each flood-mitigation scenario envisioned, which expresses both the flood volume mitigated by a particular flood-mitigation measure, its cost, its cost per percent mitigated, and the overall costs. When an accumulation of flood-mitigation measures captures the entire FEV, the FEV is essentially reduced to zero. Building higher flood-defence walls in a city at or just above the maximum river level to be mitigated does, for example, reduce the FEV to zero in one fell swoop. However, building high walls around a river in a city is only one type of flood-mitigation scenario, one that may be undesirable, so in general flood-mitigation scenarios, expressed visually in square-lake graphs, will consist of an accumulation of measures such as river-bed widening, i.e. giving-room-to-the-river (GRR), active flood-storage plains, higher flood walls and NBS. Each measure cuts a certain fraction as a rectangular or quadrilateral strip off the square flood lake, with accompanying costs displayed. The graphical cost-effectiveness analysis has been developed in a series of papers by Bokhove et al. (2018a, b, c, 2019), for several extreme river floods in the UK and France in order to facilitate and improve evidence-based decision-making by city councils and citizens' groups.

We further note that Wetropolis has inspired a project on tangible models for education and water management in Enschede, The Netherlands, involving a consortium of Dutch SMEs, schools and universities –see https://www.wetropolis.nl. It consists of parallel activities with an overarching Wetropolis theme, including:

– development of educational content and tools to raise awareness in primary schools and extend the Dutch GRR-programme in secondary school levels, cf. §5.1;

– build-up of "citizen-sensing" experiments, experiments to measure climate-related indicators such as groundwater, drought, and temperature in urban areas; it is a relatively recent form of community-based participatory environmental monitoring with success in the Netherlands, Spain, and Kosovo, cf. Woods et al. (2016); and,

– development of outreach models on the water cycle, drought, and heat stress phenomena in the local environment suitable for hands-on exploration in public settings such as museum exhibits.

## Appendix A: Numerical discretisation of entire system

Allowing for irregular time steps $\Delta t_n = t^{n+1} - t^n$ with $t^0 = 0$, the entire system (14) has the following space-time discretisation, using regular finite differences for the groundwater equation, a first-order finite-volume method with upwinding for the river equation, and a first-order forward-Euler time discretisation for all differential equations involved, as follows (cf. Morton

and Mayers (2005); Leveque (1990))

and Mayers (2005); Leveque (1990))

$$\text{River:} \quad \frac{(h_k^{n+1/2} - h_k^n)}{\Delta t_n} + \frac{(Q_{k+1/2}^n - Q_{k-1/2}^n)}{\Delta x_k} = \frac{Q_m^n}{w_r}\delta_{km} + \frac{Q_{res}^n}{w_r}\delta_{kr} + \frac{Q_{1c}^n}{w_r}\delta_{k1} \quad \text{for} \quad k = 1,\dots,N_x$$

$$\text{with} \quad Q_{k+1/2}^n = h_k^n R(h_k^n)^{2/3}\frac{\sqrt{-\partial_s b}}{C_m}, \quad Q_{1/2}^n = Q_0^n = Q_0(t^n), \quad h_k^0 = h_{0k} \tag{A1a}$$

$$\text{Moor:} \quad \frac{(h_j^{n+1} - h_j^n)}{\Delta t_n} - \frac{\alpha g}{\Delta y^2}\left(h_{j+1/2}^n(h_{j+1}^n - h_j^n) - h_{j-1/2}^n(h_j^n - h_{j-1}^n)\right) = \frac{R_m^n}{m_{por}\sigma_e} \quad \text{for} \quad j = 1,\dots,N_y - 1$$

$$\text{on} \quad y_j = j\Delta y \quad \text{with} \quad (h_{N_y}^n - h_{(N_y-1)}^n) = 0, \quad h_0^n = h_{3c}^n, \quad h_j^0 = h_{0j} \tag{A1b}$$

$$\text{Reservoir:} \quad w_{res}L_{res}\frac{(h_{res}^{n+1} - h_{res}^n)}{\Delta t_n} = w_{res}L_{res}R_{res}^n - Q_{res}^n \tag{A1c}$$

$$\text{Canal--1:} \quad w_c(L_{1c} - L_{2c})\frac{(h_{1c}^{n+1} - h_{1c}^n)}{\Delta t_n} = Q_{2c}^n - Q_{1c}^n \tag{A1d}$$

$$\text{Canal--2:} \quad w_c(L_{2c} - L_{3c})\frac{(h_{2c}^{n+1} - h_{2c}^n)}{\Delta t_n} = Q_{3c}^n - Q_{2c}^n \tag{A1e}$$

$$\text{Canal--3:} \quad w_c L_{3c}\frac{(h_{3c}^{n+1} - h_{3c}^n)}{\Delta t} = \gamma Q_{tm}^n - Q_{3c}^n \tag{A1f}$$

with

$$Q_{1c}^n = C_f\sqrt{g}\,w_c\max(h_{1c}^n - P_{1w},0)^{3/2} \quad \text{and} \quad Q_{2c}^n = C_f\sqrt{g}\,w_c\max(h_{2c}^n - P_{2w},0)^{3/2} \tag{A1g}$$

$$Q_{3c}^n = C_f\sqrt{g}\,w_c\max(h_{3c}^n - P_{3w},0)^{3/2} \quad \text{and} \quad Q_m^n = (1-\gamma)Q_{tm} \equiv \frac{(1-\gamma)}{2}m_{por}\sigma_e w_v\alpha g\frac{((h_1^n)^2 - (h_{3c}^n)^2)}{\Delta y} \tag{A1h}$$

$$Q_{res}^n = C_f\sqrt{g}\,w_{res}\max(h_{res}^n - P_{wr},0)^{3/2} \quad \text{and} \quad R(h) = w_r h/(2h + w_r), \tag{A1i}$$

in which $N_y$ is the number of regular grid points in the moor across $L_y$, $\Delta y = L_y/N_y$ and $h_j^n = h_m(j\Delta y, t^n)$ and $h_{j+1/2} = (h_{j+1} + h_j)/2$; $N_x$ for $k = 1,\dots,N_x$ the number of finite-volume cells $\Delta x_k = s_{k+1/2} - s_{k-1/2}$ for the river with cell faces $s_{k\pm1/2}$ and cell average

$$\Delta x_k h_k^n = \int_{s_{k-1/2}}^{s_{k+1/2}} h(s,t^n)\,\mathrm{d}s, \tag{A2}$$

and in which the Kronecker delta stymbol $\delta_{km} = 1$ for the cell $k$ in which $s_m$ resides and is zero elsewhere and likewise $\delta_{kr} = 1$ in the cell in which $s_{res}$ resides, etc. A stable, explicit time step is determined using suitable CFL conditions based on the information speed and the nonlinear diffusion.

## Appendix B: Wetropolis' design details

A GitHub site contains information on the materials used, building instructions as well as a historical timeline of its construction[8]. Some design tools and materials are briefly outlined as follows:

---

[8]https://github.com/obokhove/wetropolis20162020

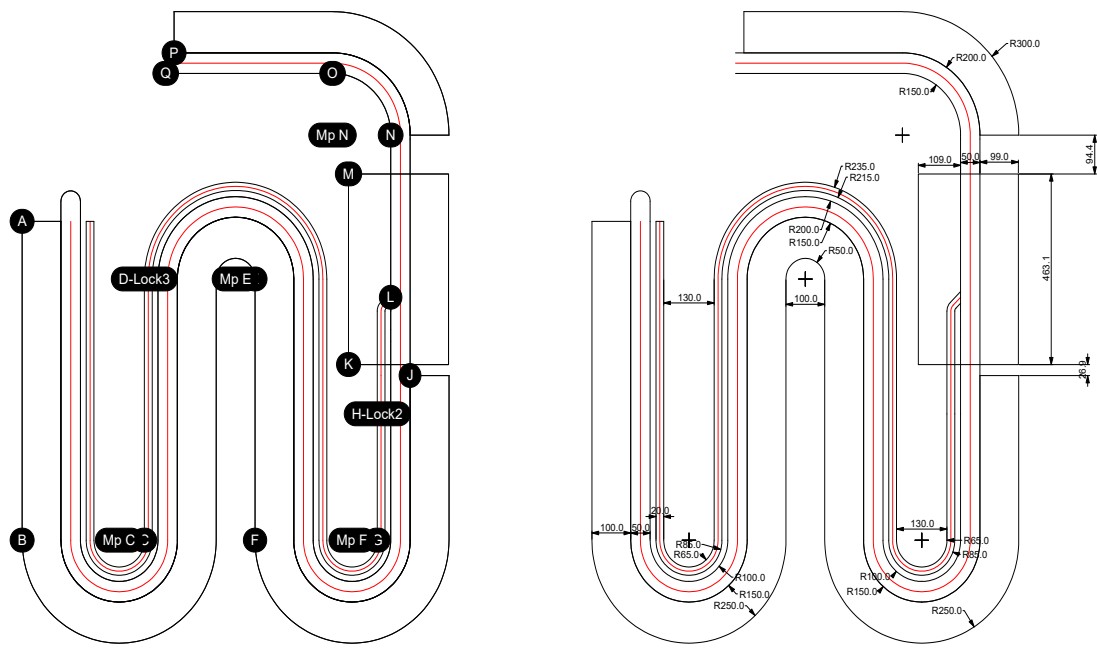

**Figure A1.** Drawings of the basic topographic landscape of Wetropolis with letter indications on the left matching coordinates on the right.

– Matlab programs of the numerical model are available on GitHub, concerning three versions. The third version "table-topt3v2019.m" was used here, which equals "tabletopt2v2016.m", except for some relabelling of figure axes.

– Blue polystyrene plates were used "Isolatieplaat polystyreen XPS" of dimensions $120 \times 60 \times 5 \text{cm}^3$ to route the terrain. Yacht varnish was mixed with fine calcinated and sieved sand and shells (with holes of $0.9$m and wire thickness of $0.1$m). The following CAD programs were used: Solidworks for the designs, saved as a Step file for import in Rhino (V5); plugin in Rhino for routing: Rhinocam 5, which generates routing/NC files; and, foams were routed on a BZT 1400 PF router/frees with a Winpc-nc driver.

– Aquarium pumps are used: Syncra 1.5, 234–240V, 50Hz, 23W, $Q{-}max = 1350$l/h, $H{-}max = 1.8$m.

– Design drawings for the topographic foam plates are given in Fig. A1 and photos of the wooden support system in Fig. B1.

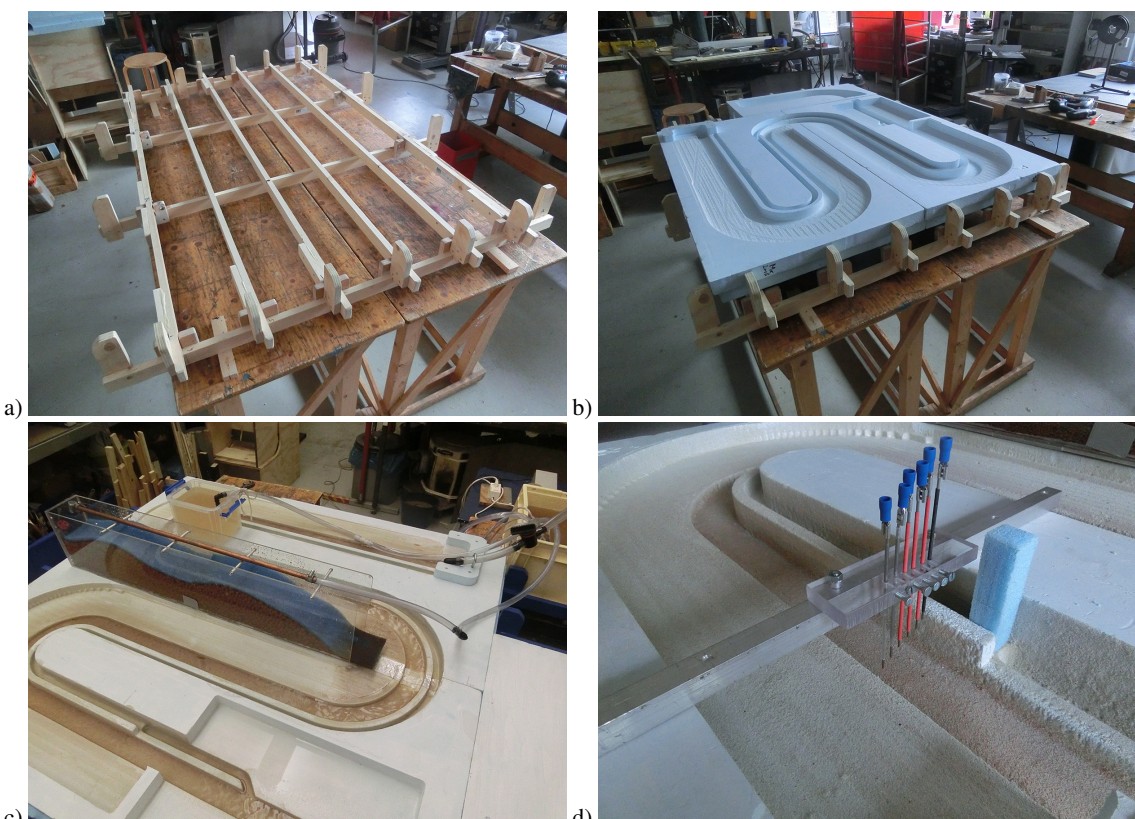

**Figure B1.** a,b) The making of the wooden support frame with its bolt-nut system. c) Overview with moor and the first reservoir; notice the aluminum tape sealing two foam plates. d) Detail of canal and sluice gate as well as a water-level measurement device involving Arduino technology.

*Acknowledgements.* The Wetropolis Flood Demonstrator originally started as an outreach project in the EPSRC UK Living with Environmental Change (LWEC) Network "Maths Foresees", under grant EP/M008525/1 for TK, OB, TH and the construction materials. The project is also partially stimulated by funding for OB and TK from the DARE project (Data Assimilation for the REsilient city) EP/P002331/1 and the local EU project "Wetropolis decision-making, participation, education and outreach with physical and augmented hydrological models" for WZ. The evaluation of Wetropolis in §4 and §5 is based on formal discussions within two workshops of the Maths Foresees network, in 2016 in Edinburgh and 2018 in Leeds, as well as feedback from the Churchtown Flood Action Group, Leeds' Armley Museum, the Newton Gateway to Mathematics and the Mathematics of Planet Earth exhibition, which (anecdotal) feedback can be found on the Wetropolis' GitHub site.

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
