# Peer review of "Wetropolis extreme rainfall and flood demonstrator: from mathematical design to outreach"

_Hydrology and Earth System Sciences, 2019_

## Referee Comment (RC1) · Anonymous Referee #1 · 22 Aug 2019

General comments:

The paper presents an innovative approach to demonstrating rainfall and flood probability and therefore discusses a topic relevant for the scope of HESS. The paper goes into detail regarding the numerical model used for the design , but lacks a presentation on the outcome of that modelling, i.e. what numerical tests were conducted and how that informed the selection of dimension, range of flows, rainfall intensities, moor and reservoir parameters that were used in the physical Wetropolis model.

Specific comments:

As noted above much of the paper focuses on presentation of the mathematical model

developed for the design of the physical model. The authors claim in the abstract that this mathematical model "is of scientific interest from a hydrodynamic modelling perspective". There have been a considerable number of mathematical models for simulation of water flow developed over the past decades and it is not immediately clear why the authors have developed their own model rather than using an existing one. It would be of interest if the authors provided a justification or an explanation. If the authors believe that they have made a contribution to mathematical modelling, they should, as a minimum, a) provide a review of relevant literature in the introduction; b) clearly specify what is novel or added value in their modelling approach; c) provide a validation of their mathematical model against observations on the physical Wetropolis model, such as by comparing the predicted and observed outflows of the system, water levels in the city, reservoir and moor, etc. Otherwise, the sections of the paper that present model components that are not related to new contribution should be shortened. The authors should present the results of the numerical tests and explain how this informed the construction of the physical model. This presentation should provide enough information for the reader to understand what were the goals of this exercise (e.g. determining which input parameters or dimensions), what are the relations between contributing flows (upstream inflow vs flows from rainfall) and conveyance (river, floodplain and canal) and in which rainfall events flooding of the city occurs. Regarding the latter, the authors should also confirm whether the predictions of the numerical model correspond to the observations on the physical model. In chapter 2.2.1, it is not clear how the floodplain accumulation and flow were modelled At the first glance and without making any calculations, it seems that canals are relatively small compared to the river and floodplain. The authors should comment on what is the role of canals in the demonstrator; does their inclusion (or omission) have other effects apart from achieving visual familiarity with the Leeds case.

Technical corrections:

on p8, line 11, symbol b usually refers to (bed) width. It seems that the authors use it for

(bed) level, in which case this word (level) should be added to avoid misunderstanding. Also for level, the symbol z could be more appropriate than b.

---

## Referee Comment (RC2) · Christopher Skinner (Referee) · 6 Sep 2019

The paper describes the mathematical design of Wetropolis, a tactile, tabletop demonstrator of flooding and probabilistic nature of the rainfall which causes it. The activity is certainly novel and innovative, and I consider the use of the Galton boards to determine rainfall events inventive with there being a lot of scope for future work into how effective this is for communication of flood likelihoods.

Unfortunately, this manuscript does not do the developed activity justice as it merely describes the mathematical model used to design and parameterise the physical model, and some description of the outreach it has been used for. In its present form it lacks

a clear research question. It is not that I don't think that either of these elements are publishable, but both need significant development to get to that stage and most likely as two separate manuscripts.

For the mathematical model, more needs to be said about how it compared to the physical model once built – how well did the maths compare to observations from Wetropolis? How does the mathematical model compare to other numerical codes of hydraulics? More guidance needs to be provided by the authors into how it can be used to advance hydrological and hydraulic modelling. As it reads at the moment I cannot see what the interest in this mathematical model would be or how anyone would utilise it.

For the outreach, there needs to be much more detail into the design choices – why was a tabletop demonstration chosen, especially when the mathematical model would lend itself to a cheaper and easier to produce numerical demonstration? What were the inspirations for the design? The design of the activity needs to be positioned within a theoretical framework for public engagement, including a relevant review of the literature.

I have no doubt that Wetropolis is a successful outreach activity, and that it is effective at achieving its aims, however, the manuscript lacks evidence for this beyond a list of events attended and some anecdotal comments. For example, on Page 21, Lines 27-29 the authors state "In particular, Wetropolis aids in raising awareness of the probabilistic character and randomness of rainfall and flooding events, also in connection with the difficulties in predicting some of these extreme events." without demonstrating how they have come to this conclusion - for example, was this through interviews or questionnaires with those partaking in the activity? The sources cited to support a further point on Line 30 also do not provide this evidence and are news articles describing that the events happened.

My recommendation for a redevelopment manuscript would be – • Clearly define the

messages they wish to be communicated and criteria they'll use to assess if they have communicated these effectively • Describe, with review of literature, why the design of Wetropolis should help them achieve this • Formal evaluation of Wetropolis at events, workshops etc against the criteria established • Discussion of how results varied between different types of events, different audiences, and different methods of communication (eg, was there an accompanying lecture)

I'm sorry I cannot provide a more positive review. I do hope the authors will revisit this and return with revised manuscripts as Wetropolis is an impressive creation and deserves to be shared widely.

Chris Skinner

---

## Referee Comment (RC3) · Anonymous Referee #3 · 6 Nov 2019

The manuscript describes an interesting outreach prototype that allows to better understand the rainfall runoff dynamic. The topic is appealing for the community and I am glad to suggest its publication. The main issue that I would like to share with the authors concerns the organization of the manuscript.

Introduction

Page 2 – lines 15-32 and page 3 lines 1-20, although interesting are off topic.

Page 4. Too many technical details about Wetropolis for an Introduction Section. I would include here eventually similar outreach prototypes and the aim of Wetropolis. What kind of experiments can be done ? which phenomena author would like to show

? Some of the info provided in the Conclusion could be added in the Intro.

Other Sections The analytical part of the paper seems in contradiction with the outreach aim. Why all these formulations are included? This should be better explained before their description otherwise could be included in the appendix. Indeed, it is not fully clear if these formulations are part of the outreach aim of they are only useful for the Wetropolis design.
* * *

---

## Author Comment (AC1) · 2 Jan 2020

**Reviewer 1 – 22[nd] August 2019**

General comments:
*The paper presents an innovative approach to demonstrating rainfall and flood probability and therefore discusses a topic relevant for the scope of HESS. The paper goes into detail regarding the numerical model used for the design, but lacks a presentation on the outcome of that modelling, i.e. what numerical tests were conducted and how that informed the selection of dimension, range of flows, rainfall intensities, moor and reservoir parameters that were used in the physical Wetropolis model.*

- To highlight the goals of the paper better, we have entirely rewritten the abstract and added extra sentences in the introduction and elsewhere, emphasizing these goals. Changes have been highlighted in red in the revised text.
- That the model "*lacks a presentation on the outcome of that modelling, i.e. what numerical tests were conducted and how that informed the selection of dimension, range of flows, rainfall intensities, moor and reservoir parameters*" is factually incorrect or a misunderstanding. The outcome of that modelling is promised in lines 2-32 of page 4 (original submission), where a range of parameters is indicated, which are determined by the mathematical and numerical modelling. On pages 15 and 16, the last part of section 2, including Fig. 5 with a simulation outcome (of the original submission), exactly what the reviewer thinks is lacking is indeed presented and was determined prior to the actual physical design could be built. Also, the detailed time-line and values used in the design process can be found on the GitHub site, to which we (did) refer, in order to check that we did not made up anything, since it contains the factual email correspondence between designer WZ & mathematician OB from 2016 onwards, as well as the original 2016 Matlab code (with an error) and the final Matlab codes used. Otherwise said, we do not understand how the misunderstanding about the perceived lack of numerical testing has arisen given that we (had) considered such testing in detail:
  - lines 20-21 on page 4 (original submission) read: "The next and crucial step in the design is to identify and determine the various unknowns in order to ascertain whether a feasible design is possible at all." This is followed by a specification of the five key unknowns ending with the statement "We chose $s_{res}$, $s_m$, $Q_0$ a priori and determined wd and $r_0$ by simulation of a simplified mathematical model", which we now modified slightly. In particular, we added the following: "… of a series of simulations of a simplified mathematical and numerical model. Note that the latter model is a lean design model exclusively geared towards obtaining quick estimates of the design parameters. It is not a model geared towards validation of measured data but a model geared towards assessing design changes quickly. After all, the goal is to design and present an outreach demonstrator not a scientific experiment even though we use up-to-date mathematical and numerical methods. While none of the elements in the mathematical and numerical model are new in separation, their holistic combination with our random rainfall is indeed novel". (Quotes from the original paper have been highlighted in blue in the revised text; new text is highlighted in red.)
  - The first sentence of section 2.3 reads "Given the choice of parameter values with (or near) the values given in Table 2, the goal is to determine a suitable rainfall speed $r_0$ and length wd via trial-and-error through numerical simulation".
  - So "*the informed selection of dimension …*", as asked for, is and was exactly done on pages 15-16 (original submission): "Via visual optimisation, i.e., monitoring when major flooding occurred in the city for the extreme or rare events of 90% rainfall in both the reservoir and moor, a suitable value of the rainfall speed is found to lie around the value $r_0 = 2.05 \cdot 10^{-4}$/s (17). The corresponding water volumes for the various Galton-board outputs required in the moor per wd are then
  $(1;2;4;8;9;18) V_{rate} = (0.18;0.36;0.72;1.44;1.62;3.24)$l/wd = $(0.018;0.036;0.072;0.144;0.162;0.324)$l/s. (18)
  "Consequently, the pump supplying the rainfall on the moor should have a maximum discharge of about 324ml/s, which is a manageable amount from a design perspective." (Correction in red of a typo.)
- Moreover, we emphasise that the model is given in detail, including the codes, such that readers

can use it to make bespoke redesigns of Wetropolis for their catchments of interest and stated that it is also a goal of our paper. Note that we have entirely rewritten the abstract and highlighted these goals now explicitly, also via further changes in the introduction.

*Specific comments:*
*As noted above much of the paper focuses on presentation of the mathematical model C1 developed for the design of the physical model. The authors claim in the abstract that this mathematical model "is of scientific interest from a hydrodynamic modelling perspective".*

- The reviewer focuses on side remark; we have removed this sentence part "*is of scientific interest from a hydrodynamic modelling perspective*" and entirely rewritten the abstract.

*There have been a considerable number of mathematical models for simulation of water flow developed over the past decades and it is not immediately clear why the authors have developed their own model rather than using an existing one. It would be of interest if the authors provided a justification or an explanation. If the authors believe that they have made a contribution to mathematical modelling, they should, as a minimum, a) provide a review of relevant literature in the introduction; b) clearly specify what is novel or added value in their modelling approach; c) provide a validation of their mathematical model against observations on the physical Wetropolis model, such as by comparing the predicted and observed outflows of the system, water levels in the city, reservoir and moor, etc. Otherwise, the sections of the paper that present model components that are not related to new contribution should be shortened.*

- We have removed that offending sentence piece and kept the mathematical and numerical modelling because it is used to "... explain this mathematical model in detail since it was a crucial step in Wetropolis' design" (old abstract). It is also key that this model is minimal, focused to guide the design; it does not and is not meant to provide a validation. We emphasise again that our manuscript is submitted to HESS 'education and communication', not as a full research article.
- In addition, at the start of section 2 we added: "While the individual modelling elements in separation are known or straightforward, their holistic combination with the statistical rain modelling as well as the subtle mass-conserving coupling between the elements is nontrivial and new. In addition, dissemination of the model is also required to facilitate adaptations by the readers. One other reason to be quite pedagogical is to reach a wider readership of enlightened and interested members of the public, including educators."
- We have added references to graduate texts by Morton and Mayers (2005) and Leveque (1990).
- Since outreach, education and communication is the purpose, validation of the model as if Wetropolis is a scientific experiment is not required. Scientific validation of Wetropolis is simply not our goal even though it could, of course, be of interest in future research, as opposed to outreach. Moreover, a validation would not change the outreach design and is as such presently irrelevant.
- There are novel components in the design model, i.e. putting all known elements together in one holistic model with its novel statistics via the Galton boards for the rainfall and the subtle connections between the different elements. But these modeling novelties are focused on delivering design guidance not validation. Moreover, for validation, a full and rather non-trivial data assimilation approach is required, and that would lead to a research paper, which is not the topic at hand.

*The authors should present the results of the numerical tests and explain how this informed the construction of the physical model. This presentation should provide enough information for the reader to understand what were the goals of this exercise (e.g. determining which input parameters or dimensions), what are the relations be- tween contributing flows (upstream inflow vs flows from rainfall) and conveyance (river, floodplain and canal) and in which rainfall events flooding of the city occurs.*

- As discussed above in the second bullet response, this is exactly what we had done. We are surprised that the reviewer missed that we exactly did use the model results for the design? See the various blue-highlighted texts, which were already present in the original manuscript.

*Regarding the latter, the authors should also confirm whether the predictions of the numerical model correspond to the observations on the physical model. In chapter 2.2.1, it is not clear how the floodplain accumulation and flow were modelled. At the first glance and without making any calculations, it seems that canals are relatively small compared to the river and floodplain.*

- We offer an alternative position, as said; our modelling results are not predictions of water height but estimates with a minimal mathematical and numerical model to predict pump strengths and several parameters for the outreach design. The model solely serves to guide the actual design at low order, not as accurate predictions. In the actual design, the use of materials and pump action was changed to some degree, which is fine given that the goal was to make an outreach flood demonstrator, not a scientific experiment, which challenge was posed to us by flood professionals. We could also have used the Navier-Stokes equations but that would have been very time consuming and impractical to estimate design changes quickly. So it is on purpose that our model is quick and minimal, which was functional, since during the design we could thus quickly change the river-channel length (see the historical time line on GitHub which demonstrates this). Moreover, the editor explicitly changed our paper from a research paper to an education and communication paper, with our agreement, thus underscoring that this is not and never was meant to be a scientific research paper. Note lines 4-9 in section 5 on page 22 (original manuscript), where the above was highlighted already: "This efficient mathematical model was first presented as a coupled system of ordinary and partial differential equations which we subsequently solved numerically to define a near-optimal design. While that mathematical model is close to a prediction model for river and groundwater flows in Wetropolis, due to its relatively minimalist nature and purpose to facilitate the design, it is likely not quite sophisticated enough to make bonafide predictions. In a final modelling step, we determined the reasonable rainfall and flow rates through numerical simulation, on which rates we based the actual design and construction of Wetropolis."
- A fundamental issue is our principle that for design purposes one-to-one agreement between design model and final demonstrator is neither required nor expected, because the end-goal is the outreach experiments and its functioning itself, not validation between observations and (design) model. Design models need to be flexible and quick since in the actual construction, adaptations leading to more efficiency naturally emerge; in the model we have rainfall percentages while adjusting pump strength is a pain so in the design we replace it by percentages per wd so the pumps simply are switched on and off. There have been a multitude of such minor design changes, which do not require a rerun of the design model.
- Floodplain accumulation has not been modelled but flooding has been modelled in a simple manner since that was adequate for the design model. This is and was clearly stated on page 14 lines 4-7 and further (original paper): "… major flooding is defined to occur when the river level significantly, i.e. by 0.01m or more, exceeds the canal--1 berm along the strip of river bordering the city plain. This is monitored visually in daily snapshots".

*The authors should comment on what is the role of canals in the demonstrator; does their inclusion (or omission) have other effects apart from achieving visual familiarity with the Leeds case.*
- This was explained, see page 16, lines 9-10 (original manuscript): "During this time major flooding is lessened, or prevented completely, because the reservoir and canal in essence act like flood-attenuation storage sites, supplying passive flood control." And page 18, lines 11-14 (original manuscript): "Hence, active flood control can be demonstrated by manually adjusting this outflow level. Outflow into the canal can be arranged separately via an adjustable valve. Note that this is slightly different from the set-up in the mathematical and numerical design model, where the outflow of moor water was partitioned between the river and canal." See also the caption of Fig. 2 of both the original and revised manuscripts. In other words, the canal also can act as minor flood alleviation. Note that we added/altered the sentences: "The river-canal

combination established is inspired by the River Aire and Leeds-Liverpool canal sharing a large part of the same river valley with the canal allowing some minor flood alleviation via (manual) flood control" and "We recall that the reservoir has valves such that the audience can store and release water interactively into the canal and river in order to control and possibly prevent flooding in the city."

Technical corrections:
*On p8, line 11, symbol b usually refers to (bed) width. It seems that the authors use it for b (bed) level, in which case this word (level) should be added to avoid misunderstanding. Also for level, the symbol z could be more appropriate than b.*

- The river bed $b=b(s)$ is a variable of the winding river coordinate s; use of the vertical coordinate z (as opposed to a variable or function) is less appropriate, according to use in many numerical modelling papers, including papers of Akers and Bokhove (2008), Ambati and Bokhove (2007ab), Tassi et al. (2007), and Rhebergen et al. (2008) in the Journal of Computational Physics. It is common to use notation $b=b(x,y)$ and, hence, here $b=b(s)$. The symbol b from "bed" is used on purpose, as opposed to the symbol z, since z is not a letter in the name "bed" and is more often the vertical coordinate. In addition, b is a well-known choice in the applied mathematical and numerical modelling community.
- However, we added a sentence to enhance clarity "Hence, the river bottom lies at $z=b(s)$ with vertical coordinate z and the river surface at $z=b(s)+h(s,t)$" in section 2.2.1.
- We removed an incorrect term in the Canal-2 equation (three times), which was a typo since the erroneous term was not present in the numerical code (which can be verified from the dated codes on the GitHub site). Similarly, the inflow of the canal into the river was missing in the write-up but was present in the codes. These corrections have been marked in red.
- We extended and modified the conclusions based on new developments and remarks by the other reviewers.

---

## Author Comment (AC2) · 2 Jan 2020

**Reviewer 2 Christopher Skinner (Referee) –6th September 2019**

*The paper describes the mathematical design of Wetropolis, a tactile, tabletop demon strator of flooding and probabilistic nature of the rainfall which causes it. The activity is certainly novel and innovative, and I consider the use of the Galton boards to deter- mine rainfall events inventive with there being a lot of scope for future work into how effective this is for communication of flood likelihoods.*

*Unfortunately, this manuscript does not do the developed activity justice as it merely describes the mathematical model used to design and parameterise the physical model, and some description of the outreach it has been used for. In its present form it lacks a clear research question.*

- Pertaining to the perceived lack of research question, we point that the challenge at hand is not a research question but rather the following: "UK flood experts therefore gave us the challenge to design an outreach tool visualising what a return period is." This article on outreach disseminates a) how we solved that challenge and b) how the tools we developed and used can lead to adaptations by the readers. We therefore disagree with the reviewer. For clarity's sake, however, we entirely rewrote the abstract and made several adaptations and changes to the introduction to explicitly state that challenge and those dissemination goals much more clearly. We also note that the editor purposely changed the article from a HESS research paper into a HESS education and communication paper suitable for this outreach project; it should therefore be reviewed as such. Changes in the text have been indicated in red.

*It is not that I don't think that either of these elements are publishable, but both need significant development to get to that stage and most likely as two separate manuscripts.*

*For the mathematical model, more needs to be said about how it compared to the physical model once built – how well did the maths compare to observations from Wetropolis? How does the mathematical model compare to other numerical codes of hydraulics?*

- We disagree for the following reasons (as we have already alluded to above):
  (i) the mathematical and numerical model constitutes a lean design model, to guide the determination of the design parameters; it is not a model meant for validation of Wetropolis seen as a scientific experiment. This was already stated in the original submission (blue highlighted original text in the revision) and is even emphasized more in the revision (new red highlighted text). Designs models must facilitate quick design changes and our model does.
  (ii) There is also a misunderstanding what such a validation would entail. Wetropolis is a portable model and at every place it is set up there are slight variations in level, pump strengths, etc. At present there are two (untested) water-level measurements possible and a camera could be placed to record the moor water level. However, a proper validation would be a research project in itself, requiring full data assimilation and parameter estimation approaches because there will be significant uncertainties in the water influx, the determination of the topography, the pump strengths, riverbed roughness, etc. That is all appropriate for a substantial research project but such an endeavour has little to do with the actual establishment of the working and successful Wetropolis flood demonstrator. Note that our outreach goal is not to show that a mathematical model can be validated with an experimental model since our outreach goal is to visualise what a return period is: we clarified this goal in the revision. In conclusion, we reject the research suggestion by the reviewer for the present outreach paper while, of course, we are in the long-term interested to augment Wetropolis with a research component –even though that was never its original intention. Our intention is and was simply to solve the above-mentioned challenge posed by flood professionals.

*More guidance needs to be provided by the authors into how it can be used to advance hydrological and hydraulic modelling. As it reads at the moment I cannot see what the interest in this mathematical model would be or how anyone would utilise it.*

- As discussed above, the model is a lean design model and has factually been used to make the design possible, which we clearly indicate with the blue-coloured text in the revised manuscript (pertaining to text already in the original submission); the model is not meant to advance hydrological and hydraulic modelling. Any reference to advances in research have been removed from the abstract, which has been rewritten entirely, and from the main text. However, what is novel about the mathematical model is the mass-conserving coupling between the various systems and its holistic overall coupling, with the random rainfall coming from the Galton board draws. This novelty was required to establish a lean model facilitating the design.
- The requested "guidance" is now given in a rewritten abstract, more clarification is now given in the introduction and selectively further in the text. These changes have been marked in red colour.

*For the outreach, there needs to be much more detail into the design choices – why was a tabletop demonstration chosen, especially when the mathematical model would lend itself to a cheaper and easier to produce numerical demonstration?*
- The reason to choose a table-top design have now been given: flood professionals had posed us that challenge and one in particular, JBA Trust, has good experience with physical models for PR and educational reasons. We now mention this explicitly, also referring to JBA's coastal wavetank, which was designed within Leeds' CDT in Fluid Dynamics, on request of JBA Trust.
- Numerical demonstrations are flat and may be uninspiring to the general public. What was requested was an interactive physical tool.

*What were the inspirations for the design?*
- The inspirations for the design were clearly given in the first few pages of the introduction, which have been further clarified with statements relating it to the design that is to follow. If that is not what is meant, the design simply popped up in OB's head and was developed in a series of iterations with designer WZ, which iteration timeline is and was available in the story told on GitHub (emails between OB and WZ literally reveal how matters developed quickly). OB and WZ are Dutch citizens and within the context of the Deltaworks, of which small-scale test versions were built in the Noord-Oost polder after the 1953 flood, conceptual modelling of river flood components like in Wetropolis is perhaps natural for Dutch engineers and designers? See also online literature of the "Waterloopkundig laboratorium" in the Noord-Oost polder.

*The design of the activity needs to be positioned within a theoretical framework for public engagement, including a relevant review of the literature.*
- We do not understand this request. Two of the authors are engineers and designers. We were given a public-engagement challenge by flood professionals, given the problems they face to explain what a return period is and how to visualise this to the general public, and we simply solved that challenge. Furthermore, this 'education and communication' paper describes in detail how we solved it. Why is that not sufficient? With theorising one does not resolve the challenge. There seems to be a collision between hands-on problem solving and theorising about solving it?
- Given the changes we have made to the manuscript, and given that Wetropolis is not going to change in any significant way when such positioning is done and given that we simply do not have the social science background to do so, we leave such a positioning to the relevant experts in a future endeavour.
- What is lacking, cf. the video made of the coastal wavetanks, is 10-15min video of the Wetropolis experience, augmenting the physical set-up, such that the visual explanation of return periods can reach a much wider audience world-wide. While making such a video is a future plan, current lack of funding (at an estimated 5000 pounds for a professional video) means it cannot be done as part of the present manuscript revision.
- The evaluation of Wetropolis in Sections 4 and 5 is and was based on formal discussions within two workshops of the Maths Foresees network, in 2016 in Edinburgh and 2018 in Leeds, as well as (new) formal feedback from the Churchtown Flood Action Group, Leeds' Armley Museum [pending] and the Turing Gateway found on the Wetropolis' GitHub site. We now made this more explicit by adding this remark in the acknowledgements and further changes within the main text.

*I have no doubt that Wetropolis is a successful outreach activity, and that it is effective at achieving its aims, however, the manuscript lacks evidence for this beyond a list of events attended and some anecdotal comments. For example, on Page 21, Lines 27-29 the authors state "In particular, Wetropolis aids in raising awareness of the probabilistic character and randomness of rainfall and flooding events, also in connection with the difficulties in predicting some of these extreme events." without demonstrating how they have come to this conclusion -for example, was this through interviews or questionnaires with those partaking in the activity?*

- The audience viewing Wetropolis and the organisers of events simply told us at the time and/or in retrospect. In addition, the reflections of a discussion session on public engagement, including Wetropolis, at EPSRC "Maths Foresees" UK network meetings have been included in this paper, see, e.g., page 21 and section 5.1 of the original submission. These sections have now been clarified.

*The sources cited to support a further point on Line 30 also do not provide this evidence and are news articles describing that the events happened. My recommendation for a redevelopment manuscript would be:*
*(a) Clearly define the messages they wish to be communicated and criteria they'll use to assess if they have communicated these effectively.*

- We have now highlighted the "messages we wish to communicate" via the rewritten abstract and additions to the introduction and throughout the text – all marked in red. The challenge posed was to create a 3D demonstrator visually explaining what a return period is with respect to flooding. We succeeded in solving that challenge and present how we solved it and how design changes can be facilitated for use by other people.

*(b) Describe, with review of literature, why the design of Wetropolis should help them achieve **this**.*

- We solve "this" as that is the challenge posed. A review of the literature is relatively futile as far as we know and as far as we have been told by various flood professionals, otherwise they would not have posed their challenge, since Wetropolis is unique in its combination of random and visualised rainfall coupled to a physical river flood model in one holistic physical model. We did refer to literature on Lego models. Even all reviewers refer to this innovative character. Why then do we need to justify that Wetropolis is innovative and solved the challenge posed while everyone already seems to agree it is innovative and solving the challenge posed?

*Formal evaluation of Wetropolis at events, workshops etc. against the criteria established.*

- The discussion on page 21 (of the first submission) is factually based on the comments we have received from the audience/viewers of Wetropolis and the formal discussions, led by Tiffany Hicks, on public engagement and Wetropolis within meetings of the UK EPSRC Maths Foresees network.
- We have now furthermore added, comments collected by the organisers of the flood exhibition at the Armley Museum [pending], the Churchtown Flood Action group workshop and the Study Group on flooding problems organised by the Turing Gateway.

*Discussion of how results varied between different types of events, different audiences, and different methods of communication (e.g., was there an accompanying lecture).*

- We did discuss how events differed already, and have added more detail about these events, and gave conclusions based on the experience at these events, which has now been clarified further; see page 21 of the original manuscript: "The strength of Wetropolis is … While Wetropolis was designed as a public outreach project, the reception from flood practitioners and scientists working in environmental fluid dynamics has been surprisingly positive; we will discuss this reception later."

*I'm sorry I cannot provide a more positive review. I do hope the authors will revisit this and return with revised manuscripts as Wetropolis is an impressive creation and deserves to be shared widely.*

---

## Author Comment (AC3) · 2 Jan 2020

**Reviewer 3**

*The manuscript describes an interesting outreach prototype that allows to better understand the rainfall runoff dynamic. The topic is appealing for the community and I am glad to suggest its publication. The main issue that I would like to share with the authors concerns the organization of the manuscript.*

*Introduction*
*1) Page 2 – lines 15-32 and page 3 lines 1-20, although interesting are off topic.*
We disagree. The first question pertains to the background on rainfall and what the public wants to know; since (random extreme) rainfall is a key component of Wetropolis, it is relevant and also is part of our motivation. The second question is relevant to understand why we made a discrete Galton board with a "discrete" tail of probability 1/16. To emphasize this aspect, we have added "… is relevant because we will create a ``discrete'' distribution with a ``rare'' tail." The third question is also relevant because it features in our design, so we have added: "We will use this direct response of floods driven by one or two days of heavy rainfall in our design". Consequently, we have now made clear how discussing the statistics of extreme events has fed into the factual Wetropolis design. Note also that reviewer 2 wanted to now what had inspired us so we now have clarified matters accordingly.

*2) Page 4. Too many technical details about Wetropolis for an Introduction Section. I would include here eventually similar outreach prototypes and the aim of Wetropolis.*
- We prefer this very simple exposition on the statistics involved, which involves back-on-the-envelop calculations with little technical difficulty, to appear here in the introduction already.

*What kind of experiments can be done?*
- Wetropolis is not a scientific experiment but a flood demonstrator. What variations can be undertaken is discussed further in section 5.1 on games. More simply, people like to trick the Galton boards with their fingers or a pen, thus triggering the extreme events they wish to take place. We further extended the discussion in 5.1 with remarks on modeling droughts and modeling climate change with suitable changes to the statistics. This can be done within the same set-up by altering the programs driving the Arduino boards.

*Which phenomena author would like to show.*
- We explain now much more clearly that there is one challenge that was posed by flood professionals and solved by us: make a 3D demonstrator that explains in a visual way what a return period is.

*3) Other Sections. The analytical part of the paper seems in contradiction with the outreach aim. Why all these formulations are included?*
- The Wetropolis construction is based on a mathematical and numerical design model. So a) without the numerical (and hence also mathematical) model there would not have been a physical Wetropolis model, and b) the design description allows other people to redesign Wetropolis and likewise adaptations to the numerical modelling may aid in their design modification. So the modelling is described in details for (re)designers. We now clearly define this as one of our goals.

*This should be better explained before their description otherwise could be included in the appendix. Indeed, it is not fully clear if these formulations are part of the outreach aim of they are only useful for the Wetropolis design.*
- The mathematical and numerical formulations have been crucial for the Wetropolis design, are de facto a design principle and they are useful in redesigns, for example by readers who wish to adapt Wetropolis to their local catchment situation, which two aspects we now have greatly emphasized. We have used this design principle, to use a lean design model to guide the experimental design, in other (outreach) fluid experiments we have made in the past, such as the bore-soliton-splash, the Hele-Shaw beach experiment and the coastal wave tank. We have highlighted to the coastal wavetank. Otherwise said, one of our theoretical foundations for outreach fluid demonstrators is to start with a lean and mathematical design model, let it guide the factual creation of the demonstrators and do not get side-tracked by detailed modelling questions because in the actual construction changes are naturally made to optimise the physical design.

---

## Author Comment (AC4) · 3 Jan 2020

There is a revised pdf for checking alongside the response; either ask the editor(s) or look (temporarily) on GitHub.

---

## Author Comment (AC5) · 3 Jan 2020

There is a revised pdf for checking alongside the response; either ask the editor(s) or look (temporarily) on GitHub.

---

## Author Comment (AC6) · 3 Jan 2020

There is a revised pdf for checking alongside the response; either ask the editor(s) or look (temporarily) on GitHub.
* * *

---

## Author Comment (AC7) · 10 Jan 2020

The response to reviewer 1 should be read together with the revision, found at the moment at (response using text in original submission indicated in blue; new text is in red): https://github.com/obokhove/wetropolis20162020/blob/master/wetropolis2019_2020.pdf

---

## Author Response (AR2)

Response to revision for "Wetropolis extreme rainfall and flood demonstrator: …"
By Bokhove et al.
For HESS, regarding hess-2019-191

Dear Editor, dear Matjaž,

Thank you very much for your comments and decision to allow minor revisions. A response to the main points raised follows below. Changes in the revised paper have been highlighted in red, with the red and blue highlighted comments in the first revision being changed to the normal black-on-white.

(i*) "I have decided to accept the paper under the condition that you really explain how you have come to the conclusion with regard to the feedback by the users and stakeholders from different workshops. It is clear that this part was not done using formal ways of getting this positive feedback (questionnaires, statistical analysis, and so forth). A scientific paper (original paper) should have had that part based on scientific methods."*

- Per (i): In section 4 after the list of exhibitions, we have now added the following: "Most of these considerations are anecdotal except for the discussions at ``Maths Foresees'' meetings and the Newton Gateway to Mathematics, which were based on formal notes of the in-depth round table and workshop discussions as well as the study-group host. We stress, therefore, that the outreach component of this study is lacking a proper scientific method (from a social-science perspective). Indeed, none of these discussions concern formal questionnaires, well-balanced questions and subsequent statistical analysis, which would constitute a formal investigation of the feedback and Wetropolis' impact. There are two reasons why such a formal analysis is lacking: the authors do not have the expertise to undertake such an analysis and, more importantly, during the showcasings for at-the-time recent flood victims we did not want to further bother these victims by conducting intrusive questionnaires in a potentially unhelpful manner. However, this is something to be considered for future demonstrations, in collaboration with the necessary experts. It also implies that the conclusions suggested below are preliminary in nature."

*"Please, take into account the suggestion from the end of the Report #1 (Christopher Skinner):"*
*(ii) "A formal evaluation isn't always required, yet the authors should make it clear that the information they use is anecdotal and based on informal feedback provided to them, and that the conclusions they make are based on this. If this is done, then this manuscript will be of interest and value to the HESS readership and would recommend it to be published – I have based my recommendation on this."*

- See our answer to point (i) above.

*(iii) "Such a "relativisation" of the positive feedback in this minor revision would give a free space for possible discussion in future on the use of such demonstrators, on pros and cons and possible misunderstandings how to apply them and what is their outreach potential. I see a possible way further and that is to apply such demonstrators in practice more in the domain of risk dialogue than working on strict mathematical validation - though, this is also needed, it may be done in future in a separate (original scientific) paper? This is a must for a wider acceptance of the Wetropolis demonstrator."*

- See our answer to point (i) above but also note that we have added a slightly more formalized feedback of a recent exhibition where Wetropolis was showcased, i.e., we added the following paragraph at the end of section 4: "Finally, we recently showcased Wetropolis (a new and improved

demonstrator based on the same design principles presented here) as part of the (biannual) Mathematics of Planet Earth exhibition organised by the corresponding Centre for Doctoral Training at Imperial College in London for about 500 to 1000 visitors over nine days, from February 15th to 23rd 2020. In line with the directives of the organisers, the audience was encouraged to volunteer bespoke feedback on two post-it boards, one for the larger exhibition and one specifically for Wetropolis. We received ten feedback posts with positive feedback as well as suggestions, the latter ranging from: (a) build more of these, (b) make an exhibition set-up for tsunamis to (c) please add a full-fledged rain and river flow predictive model of Wetropolis and compare the two [footnote]{This feedback and the feedback of all exhibitions listed above are found at https://github.com/obokhove/wetropolis20162020/tree/master/feedback .} While this feedback is still lacking an in-depth statistical basis, given that the organisers felt that a formal questionnaire would be too intrusive, it does provide additional insights into the audience' perception of Wetropolis".

(iv) *"I agree that the manuscript was transferred from Original Research Paper to Education and communication paper, this decision is from my point of view correct and opens a possibility for the manuscript to be accepted and published. You have done a great job by revising the manuscript. What stays open and is a problem is the judgment/impression that you gave at the end of the paper about the stakeholders' acceptance of the demonstrator. Putting aside the need for a strict validation of the mathematical model used for putting together the demonstrator, there is a need in a journal like HESS, that the whole research work is published in such a way that anyone can repeat it."*

- Note that given the GitHub site and the article combined, construction of Wetropolis is fully repeatable and adaptations are straightforward. In fact, Wetropolis II demonstrates this.

(v) *"Since for the outreach component no proper scientific method was presented, this is definitely a weak point, even in the category Education and communication papers. Please, state clear how you evaluate these positive feedback from different workshops and (general public). Definitely, I agree that manuscript is of interest and will generate some debate about the use of such demonstrators to raise public awareness and generally society resilience. It is only that HESS is maybe not the best journal for this more "social" part of your manuscript, even though the journals seeks for interdisciplinary papers."*

- We hope and believe that the answers to points (i)-(iv) above and the corresponding changes in the manuscript have now addressed your concerns. Please let us know if further adaptations are required.

(vi) In addition, we have made some small tuning corrections to the text, also highlighted in red, and also added the remark Chris Skinner liked in our response pertaining to our motivation (i.e., the part about the 1953 floods) in section 1. We have also added some remarks on Wetropolis II. All of these minor changes have been clearly marked in red throughout the text.

We hope that the above sufficiently addresses all concerns raised.

Kind regards and, most importantly, stay healthy,

Onno Bokhove pp. the co-authors.